# Partitioning and aggregating cross-tissue and tissue-specific genetic effects to identify gene-trait associations

Shuang Song [1], Lijun Wang [2], Lin Hou [1,3] ✉ & Jun S. Liu [4] ✉

TWAS have shown great promise in extending GWAS loci to a functional understanding of disease mechanisms. In an effort to fully unleash the TWAS and GWAS information, we propose MTWAS, a statistical framework that partitions and aggregates cross-tissue and tissue-specific genetic effects in identifying gene-trait associations. We introduce a non-parametric imputation strategy to augment the inaccessible tissues, accommodating complex interactions and non-linear expression data structures across various tissues. We further classify eQTLs into cross-tissue eQTLs and tissue-specific eQTLs via a stepwise procedure based on the extended Bayesian information criterion, which is consistent under high-dimensional settings. We show that MTWAS significantly improves the prediction accuracy across all 47 tissues of the GTEx dataset, compared with other single-tissue and multi-tissue methods, such as PrediXcan, TIGAR, and UTMOST. Applying MTWAS to the DICE and OneK1K datasets with bulk and single-cell RNA sequencing data on immune cell types showcases consistent improvements in prediction accuracy. MTWAS also identifies more predictable genes, and the improvement can be replicated with independent studies. We apply MTWAS to 84 UK Biobank GWAS studies, which provides insights into disease etiology.

Genome-wide association studies (GWAS) have identified tens of thousands of susceptibility loci for complex diseases, bringing insights into disease etiologies[1]. However, it remains a challenge to understand and interpret the GWAS findings[2,3], especially those significant hits in noncoding regions. One hypothesis is that some noncoding variants influence traits through the regulation of gene expression[4]. However, to identify the path from genomic variants to gene expression is challenging. A simple strategy is to assign the associated variant a causal link to its nearest gene, but a shorter physical distance does not necessarily indicate a closer functional connection. A counterexample is that the obesity-associated single nucleotide polymorphisms (SNPs) within *FTO* form long-range functional connections with *IRX3*[2,5]. In recent years, some large-scale consortiums have provided rich resources to identify expression quantitative trait loci (eQTLs) across various human tissues. For example, version 8 of the Genotype-Tissue Expression (GTEx v8) project collects a comprehensive set of 54 tissues from hundreds of donors[6]. Integrating eQTLs with GWAS studies reveals that a large proportion of phenotype variability in disease risk can be explained by variants that regulate the expression levels of genes[7].

Recently, transcriptome-wide association studies (TWAS) have provided a successful path to bridge SNPs, gene expressions, and complex phenotypes. TWAS includes two stages: the first stage builds a linear model to predict genetically regulated gene expression from the genotype information, and the second stage associates complex diseases with the predicted gene expression. For simplicity, we refer to the two stages as the "prediction stage" and "association stage" throughout the paper. Widely used TWAS methods include

[1]Center for Statistical Science, Department of Industrial Engineering, Tsinghua University, Beijing, China. [2]Department of Biostatistics, Yale School of Public Health, New Haven, CT, USA. [3]MOE Key Laboratory of Bioinformatics, School of Life Sciences, Tsinghua University, Beijing, China. [4]Department of Statistics, Harvard University, Cambridge, MA, USA. ✉e-mail: houl@tsinghua.edu.cn; jliu@stat.harvard.edu

PrediXcan[2], which trains an elastic net model in the prediction stage, and FUSION[8], which includes more models such as top eQTL, LASSO, and Bayesian sparse linear models. A recently proposed method, TIGAR, utilizes a data-driven nonparametric prior for the eQTL effect sizes in the prediction stage, and estimates with a latent Dirichlet process regression model[9,10]. The identified gene-trait associations shed light on the genetic bases of complex diseases. However, the limited sample sizes of eQTL studies have become a bottleneck in TWAS, resulting in low power in both prediction and association stages. In the GTEx v8 dataset, for example, the sample sizes for 21 out of the 54 tissues with genotypic information are smaller than 200.

Given the tissue-dependent nature of transcription regulation and the presence of shared eQTLs across various tissues, a joint modeling approach incorporating multiple tissues can potentially enhance the performance of TWAS. There have been some methods for improving statistical power in the association stage, including MultiXcan, which tests the joint effects of gene expression variation from different tissues[11]; and sparse-canonical-correlation-analysis-TWAS, which combines the predicted expression of multiple tissues in TWAS[12]. As for the prediction stage, the recently developed UTMOST method formulates cross-tissue expression prediction as a penalized multivariate regression problem, and introduces a group-lasso penalty on the cross-tissue effects and encourages the presence of eQTLs shared across tissues[13]. Despite the improved prediction accuracy compared with single-tissue prediction, we note that the penalty emphasizes that the effects of eQTLs are shared across all tissues, but does not distinguish between biologically related or irrelevant tissues. In other words, the method tends to only identify eQTLs shared by all tissues. However, the regulatory effect of a SNP may present in only subsets of tissues (e.g., brain-related tissues), sometimes even just one tissue (e.g., testis)[14]. Furthermore, tissue-specific eQTLs can provide a more focused mechanistic interpretation for GWAS associations than eQTLs shared across all tissues[15,16].

In this manuscript, we develop MTWAS, a flexible method to aggregate multiple tissue information for TWAS analysis. The method partitions and aggregates both cross-tissue and tissue-specific genetic effects. Considering the inherent characteristics of the genetic data, where the transcriptome of one tissue can exhibit strong correlations with others, we utilize a nonparametric missing value imputation method for inaccessible tissues. Thus, MTWAS allows for complex interactions and nonlinear data structures across tissues. In addition, we employ a stepwise procedure to select eQTLs and train prediction models by minimizing the extended Bayesian information criteria (EBIC)[17,18].

We show that MTWAS significantly improves the prediction accuracy of genetically regulated gene expression over existing methods, across all available tissues and cell types in the GTEx v8, DICE, and OneK1K datasets, and the prediction weights can then be used for identifying gene-trait associations with either individual-level genotype data or GWAS summary statistics.

## Results

### Model overview

The prediction stage of MTWAS has three steps (Fig. 1). The first step imputes values for missing entries in the sample-by-tissue expression matrix of each gene via a nonparametric imputation procedure[19]. Specifically, for each column of the matrix, we impute its missing entries by leveraging the information from other columns corresponding to relevant tissues. The imputed expression matrices are subsequently used for the identification of eQTLs in the following steps. We note that this imputation step does not involve the genotype data.

The second step detects cross-tissue eQTLs (ct-eQTLs), i.e., genetic variations that are associated with gene expression in multiple tissues. For each gene, we extract the first few principal components

(PCs) from its imputed sample-by-tissue expression matrix. Note that the first PC typically characterizes overall cross-tissue expression patterns (Supplementary Figs. 1–3), whereas others likely reflect differences across subsets of tissues (e.g., brain-related tissues versus others). The number of PCs selected is determined by magnitudes of eigenvalues of the correlation matrix, for which 2.0 is set as a default cutoff (on average 5 PCs have been chosen in the GTEx studies). Then, we regress each selected PC (can be thought of the expression vector for a "super-tissue") against the individuals' genotypes of the cis-SNPs and select ct-eQTLs based on EBIC (Methods). We show in Supplementary Fig. 4 that the performance of our method is robust to the number of PCs selected.

The third step identifies tissue-specific eQTLs (ts-eQTLs, i.e., genetic variations associated with gene expression within a particular tissue after accounting for the effects of ct-eQTLs) by the stepwise sparse linear regression method SODA[18]. Specifically, with the tissue-specific gene expression as the response and genotypes of the cis-SNP as the predictors in a linear model, SODA selects ts-eQTLs by minimizing EBIC (after accounting for ct-eQTLs). Effects of ct-eQTLs and ts-eQTLs are estimated using a weighted least squares method ("Methods"), and further utilized to infer gene-trait associations.

In the association stage, for a trait of interest, we perform an association test in a tissue-specific manner to retain the tissue specificity of gene-trait associations. We derive an explicit form of the MTWAS statistics with estimated effects of ct-eQTLs and ts-eQTLs, the GWAS summary statistics, and the reference LD matrix ("Methods").

### Improvements in prediction accuracy on GTEx tissues

We evaluate MTWAS in comparison with a few state-of-the-art methods including PrediXcan, TIGAR, and UTMOST with the GTEx v8 dataset. After quality control, 47 tissues were retained with sample sizes larger than 100 ("Methods"). The goal is to predict each individual's gene expression in these tissues from his/her genotype information. The prediction accuracy is assessed by fivefold cross-validation (CV). Compared with PrediXcan, MTWAS achieved an average improvement in prediction $R^2$ of 0.02 (SD = 0.007) across 47 tissues, which amounts to about 47.4% (SD = 17.3%) improvements (Fig. 2). In addition, MTWAS showed an average increase in prediction $R^2$ of 40.1% (SD = 25.1%) and 9.2% (SD = 2.1%) over TIGAR and UTMOST, respectively.

Figure 2a shows the improvements of the prediction $R^2$ over that of PrediXcan for all methods (0 for no improvement; negative values mean that the average prediction $R^2$ is smaller than that of PrediXcan). We note that the improvement of our method is more significant in tissues with smaller sample sizes. For example, MTWAS improves the prediction $R^2$ by an average of 60.9% over that of PrediXcan for tissues with sample sizes smaller than 200; and by 42.2% for tissues with sample sizes between 200 and 400; and by 26.9% for tissues with sample sizes larger than 400. UTMOST also outperformed PrediXcan significantly, but was inferior to MTWAS uniformly. TIGAR performed comparably to UTMOST in tissues with sample sizes larger than 400, but performed worse in tissues with smaller sample sizes. This can be explained by our intuition that cross-tissue information helps improve the prediction accuracy, especially in tissues with small sample sizes. We also provide the average signed prediction $R^2$, calculated as the product of Pearson's correlation coefficient sign and the $R^2$ value (Table 1). Given that the signed prediction $R^2$ can be negative, its average tends to be lower and more conservative compared with the prediction $R^2$ for all methods. Nevertheless, MTWAS still showed the best performance in terms of the signed prediction $R^2$.

MTWAS's improvement in prediction accuracy can be attributed to three aspects: (i) the imputation for unobserved entries in expression matrices utilizes cross-tissue information and increases effective sample sizes of training datasets; (ii) while cross-tissue effects aggregate information from various tissues, tissue-specific effects retain

**Step 1: Imputing missing entries**

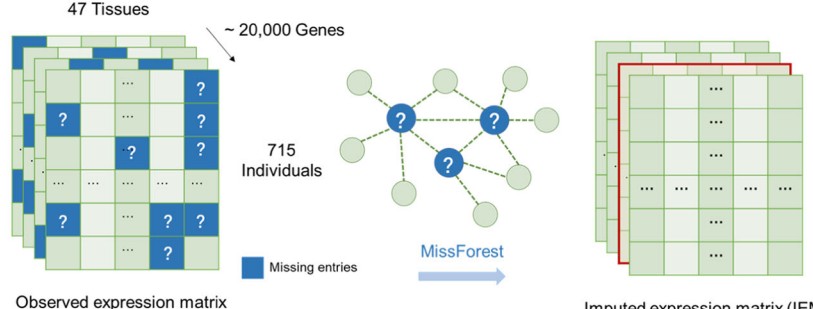

**Step 2: Detecting cross-tissue eQTLs**

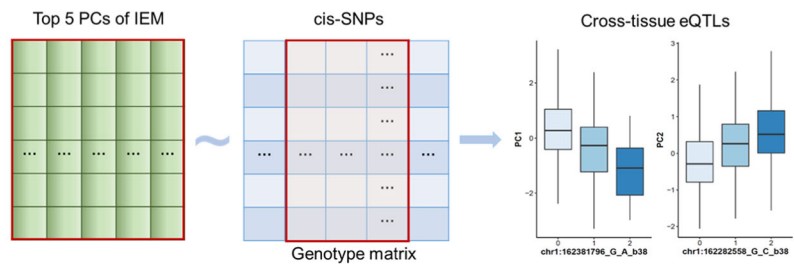

**Step 3: Estimating cross-tissue and tissue-specific effects**

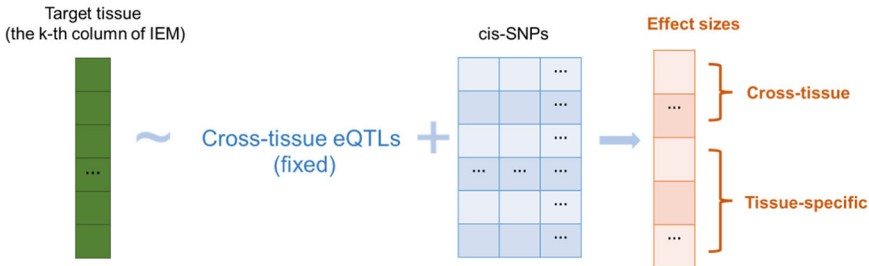

**Fig. 1 | Schematic diagram of the prediction stage of MTWAS.** The prediction stage of MTWAS is divided into three steps. Step 1 does the expression imputation, where the missing entries of an observed expression matrix are imputed using information from other relevant tissues with the algorithm MissForest. Step 2 identifies cross-tissue eQTLs; and Step 3 detects tissue-specific eQTLs, and estimates effect sizes.

their inherent tissue-specific characteristics, and both contribute to the prediction; (iii) although computationally more demanding, EBIC for linear and generalized linear regression models under high-dimensional settings appears to be superior to other variable selection criteria such as those used in lasso and elastic net[18].

We performed more experiments to further demonstrate each of the aforementioned points. For point (i), we applied PrediXcan, TIGAR, and UTMOST to the imputed expression matrix (denoted as PrediXcan-imp, TIGAR-imp, and UTMOST-imp, respectively), and compared the results with those obtained by the respective original methods. Across the 47 tissues, all methods performed better with our imputed data than with the original unimputed data (Fig. 2b, c and Supplementary Fig. 5). For example, PrediXcan-imp improved the prediction $R^2$ over PrediXcan by a minimum of 5.32% in muscle skeletal (sample size = 601) to a maximum of 32.7% in the uterus (sample size = 107).

For point (ii), instead of partitioning eQTLs into ct-eQTLs and ts-eQTLs, we directly performed regression with all cis-SNPs as predictors on each tissue, and selected variables using EBIC. We denoted this method as MTWAS-tissue. As shown in Fig. 2b, c and Supplementary Fig. 5, MTWAS achieved higher prediction $R^2$ compared with MTWAS-tissue for all 47 tissues, suggesting that our partition and aggregation of ct-eQTLs and ts-eQTLs indeed helped. For point (iii), we compared

the performances of MTWAS-tissue, PrediXcan-imp, TIGAR-imp, and UTMOST-imp. We found that MTWAS-tissue outperformed the other three methods across all tissues. As all the tested methods use imputed gene expression as input, the advantage of MTWAS-tissue is solely attributed to the use of EBIC.

We consider two criteria for calling a gene predictable for a method: a common criterion (with $R^2 > 0.01$) and a more stringent one (with false discovery rate (FDR) <0.05). MTWAS identified more predictable genes than the other three methods under both criteria (Table 1, Fig. 3, and Supplementary Fig. 6). The numbers of predictable genes identified by MTWAS ranged from 10,444 (for whole blood) to 17,052 (for uterus) under the common criterion, and from 2451 (for vagina) to 8229 (for nerve tibial) under the stringent criterion. Compared to PrediXcan and UTMOST, MTWAS identified an average of 41.6% (SD = 10.0%) and 2.6% (SD = 1.3%) more predictable genes under the common criterion, and 92.0% (SD = 48.6%) and 19.2% (SD = 4.0%) more under the stringent criterion, respectively.

In addition to the GTEx analysis, we also performed an independent replication study by applying the weights trained with the GTEx samples for Epstein–Barr virus (EBV) transformed lymphocytes to the expression levels of the 373 European individuals from the GEUVADIS lymphoblastoid cell lines (LCLs)[20]. MTWAS also achieved a better prediction $R^2$ (Table 2) compared to PrediXcan, TIGAR, and UTMOST.

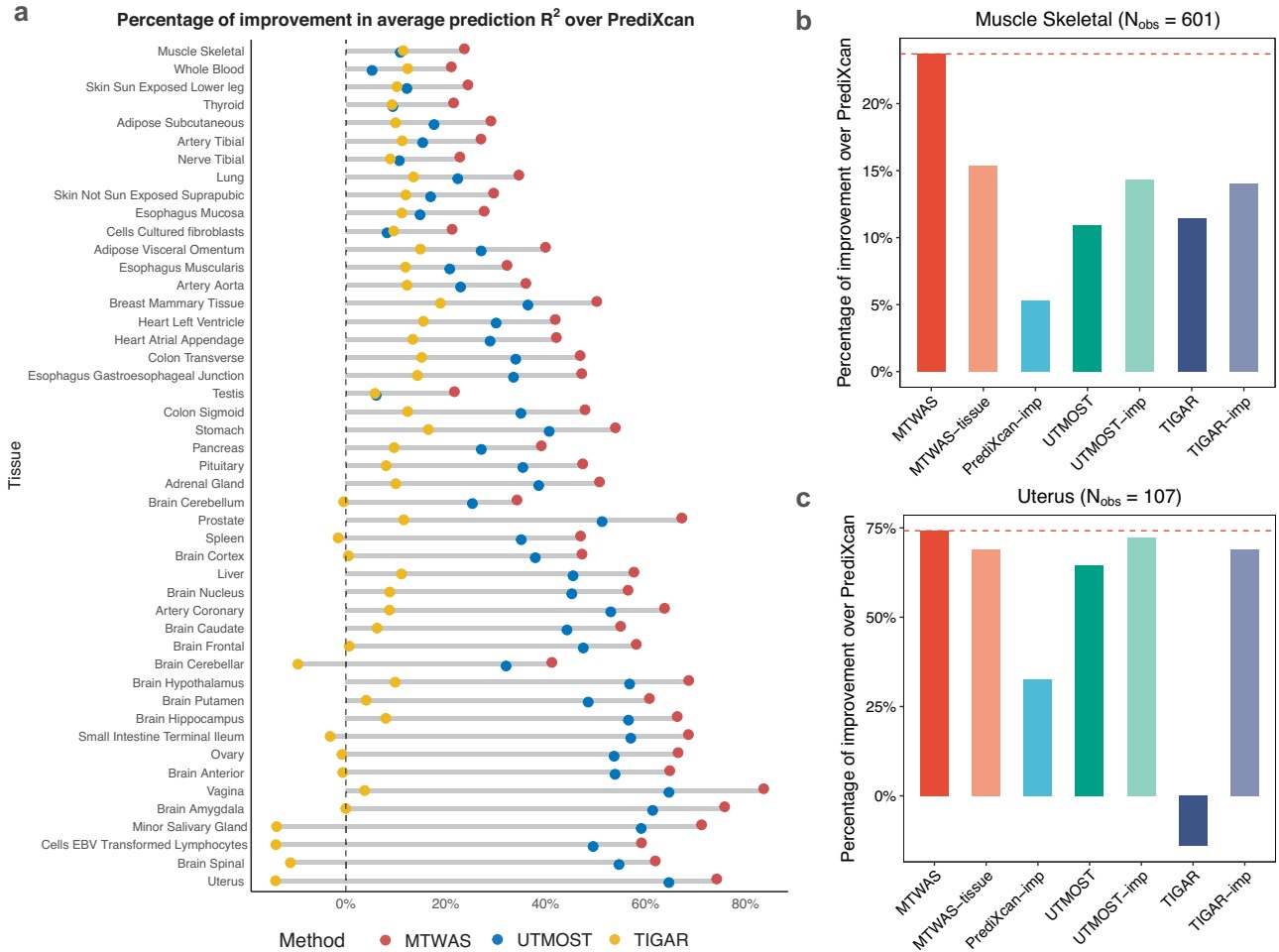

**Fig. 2 | The prediction $R^2$ evaluated in the GTEx datasets. a** Improvements of the prediction $R^2$ (average of the fivefolds in the fivefold CV) over PrediXcan, of MTWAS, UTMOST, and TIGAR in the 47 tissues. **b** Improvements of the prediction $R^2$ over PrediXcan of gene expressions of the muscle skeletal, which has the largest sample size in the GTEx datasets. **c** Improvements of the prediction $R^2$ over PrediXcan of gene expressions of the uterus, which has the smallest sample size in the GTEx datasets. MTWAS, MTWAS-tissue, PrediXcan-imp, UTMOST-imp, and TIGAR-imp were trained with the imputed data. The red dashed line marks the performance of MTWAS. $N_{obs}$ is the sample size. Source data are provided as a Source Data file.

**Table 1 | The prediction accuracy of MTWAS, PrediXcan, UTMOST, and TIGAR**

|  | MTWAS | PrediXcan | UTMOST | TIGAR |
|---|---|---|---|---|
| Average prediction $R^2$ | 0.061 | 0.041 | 0.056 | 0.044 |
| Average signed prediction $R^2$ | 0.047 | 0.031 | 0.041 | 0.030 |
| # predictable genes ($R^2 > 0.01$) | 15,066 | 10,761 | 14,703 | 13,699 |
| # predictable genes ($FDR < 0.05$) | 5027 | 2959 | 4244 | 3358 |

The results are based on the average over 47 GTEx tissues.

The prediction $R^2$ was improved by 47.8%, 126.7%, and 6.3%, respectively. MTWAS achieved consistently higher prediction $R^2$ than the other three methods in different quantiles ($P < 2.2 \times 10^{-16}$; one-sided Kolmogorov–Smirnov test). MTWAS also identified more predictable genes with both the common and stringent criteria (Table 2).

### Improvements in prediction accuracy on immune cell types

We applied our method in two eQTL datasets of multiple cell types. The DICE dataset includes the bulk RNA-seq data of 91 samples on 13 types of immune cells and 2 activation conditions[21]. We omitted the results of TIGAR because the inadequate sample size may lead to an over-fitted model for the method[10]. MTWAS showed consistent improvement of prediction accuracy compared with PrediXcan and UTMOST, with average improvements of 77.69% (SD = 2.09%) and 5.87% (SD = 1.17%) in prediction $R^2$ (Supplementary Fig. 7a). MTWAS also identified about 5 times as many predictable genes as PrediXcan, and about 14 times as UTMOST in total (Supplementary Fig. 7b).

Our method is also applicable to single-cell RNA-seq (scRNA-seq) data. We generated pseudo-bulk data matrices comprising 14 immune cell types from the OneK1K dataset, which consist of 1.27 million peripheral blood mononuclear cells collected from 982 donors[22]. Consistent with the results obtained from the bulk data, we found that MTWAS achieved the highest prediction $R^2$, along with identifying the largest number of predictable genes across all cell types (Supplementary Fig. 8). Specifically, MTWAS identified from 84 (CD4 SOX4 cells, $N_{cell} = 4065$) to 3665 (NK cells, $N_{cell} = 463,528$) predictable genes under the stringent criterion, with an improvement of 104.8% (SD = 61.6%) and 73.3% (SD = 12.1%) compared with PrediXcan and UTMOST, respectively. We note that the improvements of multi-cell-type TWAS methods (both MTWAS and UTMOST) over the single-cell-type method (PrediXcan) are more significant in cell types with fewer cell counts and thus lower statistical power in identifying predictable

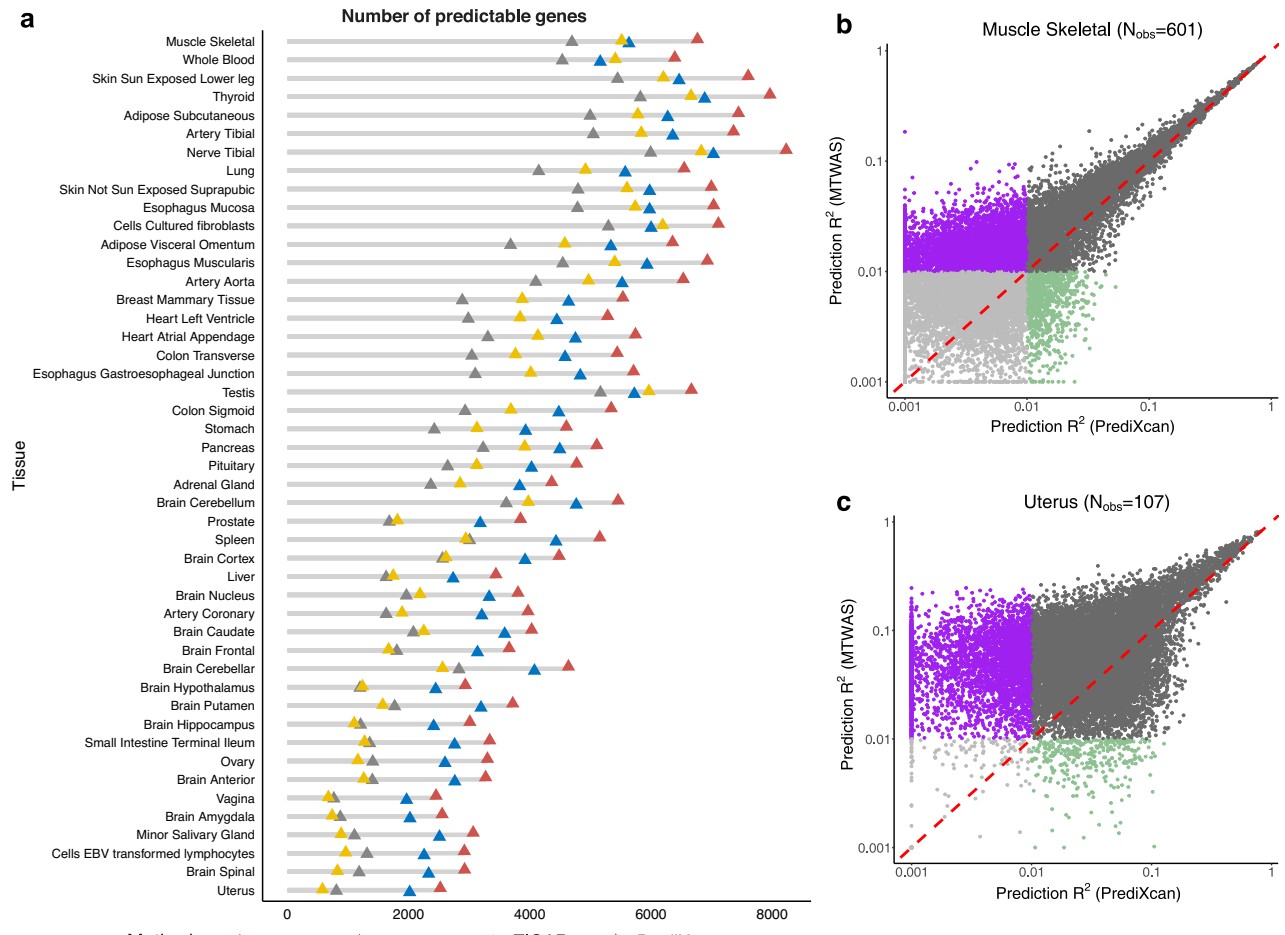

**Fig. 3 | Predictable genes in the GTEx datasets. a** Numbers of predictable genes of MTWAS, UTMOST, PrediXcan, and TIGAR in the 47 tissues, under the threshold FDR < 0.05. Tissues are arranged in descending order of their respective sample sizes. **b** Comparison between numbers of genes with prediction $R^2 > 0.01$ for MTWAS and PrediXcan in muscle skeletal, which has the largest sample size among the GTEx datasets. **c** Comparison between numbers of genes with prediction $R^2 > 0.01$ for MTWAS and PrediXcan in uterus, which has the smallest sample size among the GTEx datasets. Purple and green dots represent genes that have prediction $R^2 > 0.01$ only for MTWAS and PrediXcan, respectively. Darker and shallower gray dots represent genes that consistently have prediction $R^2 > 0.01$ and $R^2 \leq 0.01$, respectively, using both methods. $N_{obs}$ is the sample size. Source data are provided as a Source Data file.

**Table 2 | Replication study on the GEUVADIS cohort for lymphoblastoid cell lines**

|  | MTWAS | PrediXcan | UTMOST | TIGAR |
|---|---|---|---|---|
| Average prediction $R^2$ | 0.034 | 0.023 | 0.032 | 0.015 |
| Average signed prediction $R^2$ | 0.031 | 0.022 | 0.030 | 0.013 |
| # predictable genes ($R^2 > 0.01$) | 5176 | 2841 | 4894 | 2196 |
| # predictable genes (FDR < 0.05) | 4339 | 2330 | 4093 | 1312 |

The training weights are based on the EBV transformed lymphocytes in the GTEx datasets.

genes. This highlights the value of leveraging cross-cell-type information to improve predictions in cell types with limited data.

**Investigation of cross-tissue and tissue-specific effects**

MTWAS partitions eQTLs into ct-eQTLs and ts-eQTLs, and aggregates their effects in prediction. We observed that both ct-eQTLs and ts-eQTLs are significantly enriched in promoter regions ($P < 2.2 \times 10^{-26}$, one-sided Fisher's exact test), and more enriched than those selected by PrediXcan and UTMOST (Fig. 4a). In addition, the combined annotation dependent depletion (CADD) scores of ct-eQTLs are also significantly higher than that of ts-eQTLs ($P = 9.4 \times 10^{-11}$, one-sided $t$ test, Fig. 4b). Higher CADD scores indicate more severe deleteriousness of the eQTLs shared across multiple tissues. CADD scores of both ct-eQTLs and ts-eQTLs identified by MTWAS are higher than those eQTLs selected by PrediXcan and UTMOST.

We further consider two subsets of the predictable genes of MTWAS according to whether the genes are regulated by ct-eQTLs only or by ts-eQTLs only, referring to as cross-tissue genes (ct-genes) and tissue-specific genes (ts-genes), respectively. Different characteristics can be observed for the two subsets. KEGG enrichment analysis demonstrates that ct-genes are enriched for lysosome ($P = 5.6 \times 10^{-4}$), peroxisome ($P = 5.6 \times 10^{-4}$), phagosome ($P = 8.8 \times 10^{-4}$), and metabolism pathways that are abundant in all tissues and cell types (Supplementary Figs. 9 and 10). In contrast, ts-genes are enriched in pathways that are critical in differentiating between cell types, as was also observed in a recent study[23].

For example, ts-genes in the brain cerebellar are enriched in the neuroactive ligand-receptor interaction pathway ($P = 5.4 \times 10^{-5}$) and the taste transduction pathway ($P = 1.7 \times 10^{-4}$). The esophagus mucosa ts-genes are enriched in the gastric acid secretion pathway

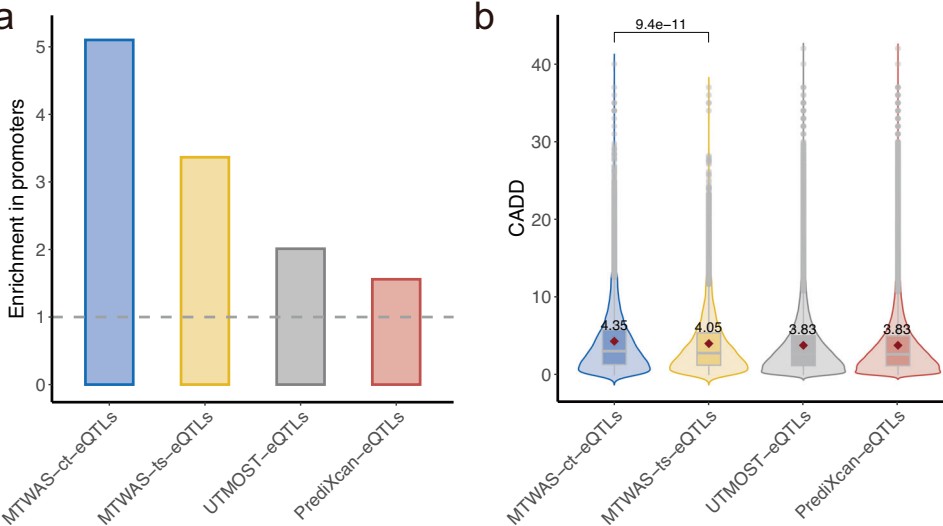

**Fig. 4 | Characterization of the eQTLs identified by MTWAS, PrediXcan, and UTMOST.** The eQTLs identified by MTWAS are partitioned into cross-tissue eQTLs and tissue-specific eQTLs. The numbers of identified eQTLs are 20,976 (MTWAS-ct), 14,072 (MTWAS-ts), 271,881 (UTMOST), and 569,512 (PrediXcan). **a** Enrichment in promoters of the eQTLs identified by TWAS methods. **b** Violin plot shows the distribution of CADD scores across four groups of identified eQTLs. The means are marked with red diamonds and labeled with the respective values for each category. The medians are marked by horizontal lines in the internal boxplots. The lower and upper hinges correspond to the 25th and 75th percentiles. Whiskers extend from the hinge to the value no further than 1.5 of the interquartile range. Data points beyond the whiskers are plotted individually. A two-sided $t$ test indicates a significant difference in CADD scores between ct-eQTLs and ts-eQTLs identified by MTWAS. Source data are provided as a Source Data file.

($P = 2.1 \times 10^{-4}$), and the cAMP signaling pathway ($P = 3.8 \times 10^{-5}$). It has been shown that esophageal mucosa expresses predominantly EP2 receptors and esophageal ulceration increases the expression of the EP2 receptor, activation of CREB, which is the downstream target of the cAMP signaling. The ts-genes in the pancreas are also enriched in multiple pathways, such as circadian entrainment ($P = 4.8 \times 10^{-5}$), protein digestion and absorption ($P = 6.8 \times 10^{-5}$), dopaminergic synapse ($P = 2.7 \times 10^{-4}$), insulin secretion ($P = 2.9 \times 10^{-4}$), morphine addiction ($P = 3.7 \times 10^{-4}$), and pancreatic secretion ($P = 6.3 \times 10^{-4}$), etc.

Furthermore, we found that ts-genes are more intolerant to protein-loss-of-function compared with ct-genes, as evaluated by LOEUF[24] and pLI[25]. Specifically, ts-genes had significantly lower LOEUF ($P = 3.2 \times 10^{-11}$, two-sided $t$ test) and higher pLI ($P = 4.1 \times 10^{-16}$, two-sided $t$ test) than ct-genes, indicating that ts-genes are more variation intolerant compared with ct-genes. This is consistent with a previous finding that ts-genes endure stronger selective pressures compared with ct-genes[23]. This has important implications for understanding genetic bases of disease, as mutations in tissue-specific genes may be more likely to lead to specific diseases affecting particular tissues or organs.

### Applications to GWAS studies on UKBB phenotypes

We applied MTWAS, UTMOST, TIGAR, and PrediXcan to 84 UKBB self-reported cancer and non-cancer illness phenotypes with effective sample sizes larger than 5000 (a detailed summary can be found in Supplementary Data 1). We use the Bonferroni correction to account for multiple testing, and report genes with $R^2 > 0.01$ in the prediction stage and adjusted $P$ value <0.05 in the association stage (Supplementary Data 2). MTWAS, UTMOST, and PrediXcan identified significant genes in 44, 34, and 39 of the 84 phenotypes, respectively. Specifically, for 8 phenotypes including heart valve problems, gastric/stomach ulcers, irritable bowel syndrome, peritonitis, muscle/soft tissue problems, iron deficiency anemia, other renal/kidney problems, and chronic fatigue syndrome, MTWAS identified genes whose expressions are significantly associated, whereas UTMOST and PrediXcan did not find any associated genes. In addition, MTWAS identified a greater number of genes than both UTMOST and PrediXcan in 20

phenotypes, while UTMOST and PrediXcan identified more genes than MTWAS in 2 and 19 phenotypes, respectively. UTMOST identified the highest number of gene-tissue association pairs in 24 phenotypes. MTWAS followed closely, identifying the majority of gene-tissue pairs in 16 phenotypes. In contrast, PrediXcan only managed to identify the most gene-tissue pairs in 3 phenotypes. We also found that MTWAS identified more gene-tissue pairs than MTWAS-tissue, showing the benefits of incorporating ct-eQTLs in the prediction model.

A Venn diagram is provided to illustrate genes that are significantly associated with heart attack/myocardial infarction (MI) in the UKBB identified by MTWAS, UTMOST, PrediXcan, and TIGAR (Supplementary Fig. 11). MTWAS identified 30 independent genes and 114 gene-tissue pairs associated with MI, among which 7 genes were not identified by UTMOST, PrediXcan, or TIGAR (Table 3). We performed KEGG pathway analysis on the ct-genes and ts-genes that are significantly associated with MI. The ct-genes are enriched in metabolism and lysosome pathways; while the ts-genes are enriched in more tissue-specific pathways (Supplementary Fig. 12).

Below we discuss biological insights for genes that are uniquely identified by MTWAS. The BRCA-1 associated protein gene, *BRAP* ($P = 1.87 \times 10^{-10}$ in artery aorta and $5.39 \times 10^{-8}$ in artery coronary), is crucial for both cardiac development and myocardial function. The absence of *BRAP*, as observed in knockout models, resulted in cell cycle arrest, diminished proliferation of cardiomyocytes, and early onset of heart failure[26]. Conversely, the introduction of transgenic *BRAP* overexpression led to an augmentation in myocardial mass and increased cell cycle activity specifically in neonatal cardiomyocytes. Given the critical role of the aorta in systemic circulation and the coronary arteries in supplying oxygen-rich blood to the myocardium, the dysregulation of *BRAP* expression in arterial tissues could disrupt the delicate equilibrium of cardiomyocyte growth and function, potentially leading to the development and progression of heart failure. Another example is that prohibitin (*PHB*) ($P = 4.66 \times 10^{-8}$ in adipose subcutaneous) has emerged as a potential therapeutic target for diabetic cardiomyopathy (DCM). In a type 2 diabetic rat model, *PHB* overexpression alleviated insulin resistance, left ventricular dysfunction, and fibrosis, suggesting its promise in treating human DCM[27]. In

**Table 3 | Independent TWAS risk genes of heart attack/myocardial infarction (MI) in the UKBB, identified by MTWAS**

| Gene | Chromosome | Most significant tissue | MTWAS $P$ value | # tissues detected | PrediXcan | UTMOST | TIGAR | Evidence |
|------|-----------|------------------------|-----------------|---------------------|-----------|--------|-------|----------|
| *AIDA* | 1 | Brain Cerebellar | $6.18 \times 10^{-11}$ | 2 | Y | Y | | 49 |
| *CELSR2* | 1 | Muscle Skeletal | $2.18 \times 10^{-9}$ | 4 | Y | Y | | 50 |
| *FAM177B* | 1 | 10 GTEx tissues* | $9.10 \times 10^{-11}$ | 10 | Y | Y | Y | 51 |
| *MIA3* | 1 | Cells Cultured Fibroblasts | $1.39 \times 10^{-13}$ | 37 | Y | Y | Y | 52 |
| *PSMA5* | 1 | Liver, Nerve Tibial | $4.77 \times 10^{-8}$ | 2 | Y | | | |
| *PSRC1* | 1 | Esophagus Muscularis | $4.56 \times 10^{-10}$ | 7 | Y | Y | Y | 50 |
| *SORT1* | 1 | Heart Left Ventricle | $2.71 \times 10^{-9}$ | 10 | Y | Y | | 53 |
| *LPA* | 6 | Liver | $5.23 \times 10^{-10}$ | 1 | Y | Y | | 54,55 |
| *PHACTR1* | 6 | Artery Tibial | $1.36 \times 10^{-13}$ | 2 | | Y | | 56 |
| *CDKN2B* | 9 | Brain Anterior, Small Intestine Terminal Ileum | $4.11 \times 10^{-29}$ | 9 | Y | Y | Y | 57 |
| *IFNA10* | 9 | Artery Aorta | $2.42 \times 10^{-16}$ | 1 | | Y | Y | |
| *IFNA13* | 9 | Brain Nucleus | $5.60 \times 10^{-9}$ | 1 | | | Y | |
| *IFNA14* | 9 | Esophagus Gastroesophageal Junction | $2.30 \times 10^{-19}$ | 1 | | Y | Y | |
| *IFNA16* | 9 | Skin Sun Exposed Lower Leg | $2.30 \times 10^{-19}$ | 3 | | Y | Y | |
| *IFNA17* | 9 | Brain Amygdala | $1.16 \times 10^{-09}$ | 1 | Y | | | |
| *IFNA2* | 9 | Brain Cerebellum | $2.30 \times 10^{-19}$ | 3 | | | Y | |
| *IFNA5* | 9 | Brain Putamen | $8.80 \times 10^{-20}$ | 2 | Y | Y | | |
| *IFNA6* | 9 | Brain Hippocampus | $1.65 \times 10^{-11}$ | 3 | | | Y | |
| *IFNA7* | 9 | Cells Cultured Fibroblasts | $8.58 \times 10^{-14}$ | 1 | | | Y | |
| *IFNA8* | 9 | Heart Left Ventricle | $8.80 \times 10^{-20}$ | 1 | | Y | Y | |
| *IFNB1* | 9 | Esophagus Mucosa | $3.06 \times 10^{-11}$ | 1 | Y | | | |
| *IFNW1* | 9 | Brain Putamen | $1.54 \times 10^{-11}$ | 1 | | Y | Y | |
| *BRAP* | 12 | Artery Aorta | $1.87 \times 10^{-10}$ | 2 | | | | 26,58,59 |
| *ERP29* | 12 | Artery Tibial | $7.12 \times 10^{-9}$ | 1 | | | | 29 |
| *PPP1CC* | 12 | Adrenal Gland | $7.12 \times 10^{-9}$ | 1 | | | | |
| *FES* | 15 | Cells Cultured Fibroblasts | $1.35 \times 10^{-8}$ | 1 | Y | Y | Y | 60 |
| *PHB* | 17 | Adipose Subcutaneous | $4.66 \times 10^{-8}$ | 1 | | | | 27 |
| *APOC1* | 19 | Brain Nucleus | $6.97 \times 10^{-9}$ | 1 | | | | 61 |
| *FOSB* | 19 | Brain Cortex | $4.04 \times 10^{-8}$ | 3 | | | | 28 |
| *MYPOP* | 19 | Cells Cultured Fibroblasts | $4.04 \times 10^{-8}$ | 1 | | | | |

The most significant tissue and MTWAS $P$ value (two-sided) denote the tissue with the highest significance level for each gene. Literature as evidence indicating the associations between the identified gene and MI is provided.
*Brain Amygdala, Brain Nucleus, Cells Cultured Fibroblasts, Cells EBV Transformed, Esophagus Gastroesophageal Junction, Liver, Lung, Minor Salivary Gland, Prostate, and Whole Blood.

addition, *FOSB* (significant in multiple tissues) has been identified as a potential diagnostic biomarker and therapeutic target for heart failure in a recent study[28]. We also notice that *ERP29* ($P = 7.12 \times 10^{-9}$ in artery tibial), involved in Connexin43 (Cx43) hemichannel assembly, plays a crucial role in Cx43 stability. Dysregulation of Cx43 is linked to myocardial diseases, and *ERP29*'s function suggests a potential connection to these conditions, providing insights into connexin-related myocardial disorders[29].

In addition, MTWAS identified that the expression of *ANKRD55* in colon transverse is significantly associated with rheumatoid arthritis ($P = 4.69 \times 10^{-8}$). *ANKRD55* encodes a protein called ankyrin repeat domain 55, which is well-established to be a potential risk factor for autoimmune diseases[30]. Another example is the association between *NOX4* and malignant melanoma in the thyroid identified by MTWAS ($P = 8.37 \times 10^{-12}$). As a regulator of glycolysis within thyroid cells, *NOX4* facilitates the proliferation of cancerous thyroid cells through the generation of mitochondrial reactive oxygen species[31]. In papillary thyroid carcinomas, the down-regulation of the sodium/iodide symporter induced by the BRAFV600E mutation is mediated by *NOX4*[32]. Our results indicate that *NOX4* could be a shared molecular link between malignant melanoma and thyroid dysfunction. In fact, emerging evidence suggests that malignant melanoma and papillary thyroid carcinoma may occur concurrently[33–35]. In addition, the hypothyroidism condition is likely to promote melanoma spread, which suggests the protective effect of thyroid hormones against disease progression[34]. Remarkably, *NOX4* inhibitors have shown promises as adjuncts to current therapies, particularly for melanoma patients with *BRAF* mutations[36]. These findings highlight the comparative advantages of MTWAS in gene and gene-tissue pair identification across multiple phenotypes, emphasizing its potential as a valuable tool in genetic research and analysis.

**Computational efficiency**

We compared the computational efficiency of MTWAS with the other multi-tissue TWAS method UTMOST. For MTWAS, we reported the CPU time for (i) imputing the missing entries of the expression matrices; and (ii) identifying eQTLs and estimating the eQTL weights, while UTMOST only involves the second process (Supplementary Table 1). Taking chromosome 1 (1911 genes) as an example, it takes MTWAS about 18 min to impute missing entries. The eQTL identification and weights estimation process take about 45 min for MTWAS, which is about half the time required by UTMOST (87 min). The computation was performed with an Intel Xeon processor with 2.90 GHz and 128 cores.

## Discussion

Although TWAS have supplemented GWAS loci with valuable insights into disease mechanisms, the accuracy of gene expression prediction remains moderate. TWAS mainly include two key steps: the prediction of genetically regulated gene expression and the association of genetically regulated gene expression with disease traits. Previous multi-tissue TWAS methods, such as UTMOST, aggregate information across multiple tissues in the prediction step by encouraging the presence of shared eQTLs across all tissues, thus improving prediction accuracy. However, when eQTLs are only shared in a subset of tissues for a gene, to encourage common effects for all tissues may not be an optimal modeling approach.

To address these limitations, we propose a statistical framework, MTWAS, which partitions and aggregates cross-tissue and tissue-specific genetic effects in predicting the genetically regulated gene expression. MTWAS first imputes the gene expression data of multiple tissues, and then employs the EBIC criterion to select ct-eQTLs and ts-eQTLs based on the imputed dataset. We have demonstrated that MTWAS outperforms existing methods in achieving higher accuracy in predicting gene expression across all tissues, and has greater power in identifying gene-trait associations. The classification of ct-eQTLs and ts-eQTLs also brings insights into the genetic regulation of gene expression.

Applications to multi-cell-type bulk and single-cell RNA-seq datasets showcase that MTWAS works well with single-cell transcriptomes. Currently, we directly applied MTWAS to pseudo-bulk data of single-cell studies. For future work, it would be interesting to develop prediction models tailored for scRNA-seq data studies.

To identify ct-eQTLs, we performed PCA analysis to the sample-by-tissue matrix for each gene to extract the cross-tissue information. We clarify that the PCA used here is different from that used for identifying batch effects. To correct for the batch effect in the gene expression dataset, we regressed out the probabilistic estimation of expression residuals (PEER) factors in data preprocessing.

It is important to acknowledge that, like other TWAS methods, the gene-trait associations identified by MTWAS do not necessarily indicate causal relationships. Instead, they serve as valuable pointers toward potential functional links between genes and diseases, warranting further investigations through experimental validations and functional studies. We note that leveraging some fine-mapping methods, such as SuSiE[37] and DAP-K[38,39] for identifying causal QTLs, can improve the prediction accuracy for certain genes (Supplementary Table 2). Therefore, it is conceptually advantageous to integrate fine-mapping methods into our cross-tissue TWAS framework to prioritize causal genes, which can be a promising future direction of our research.

In conclusion, the MTWAS framework improves the prediction accuracy of gene expression and the statistical power for identifying gene-trait associations over existing TWAS methods. We believe that MTWAS is a valuable tool in deciphering the genetic bases of complex traits and facilitating personalized treatment strategies.

## Methods

### Imputation of gene expression data

The first step of our method is to impute expression data. Genotype information is not used at this step. For each gene, we consider its expression matrix containing $N$ samples and $K$ tissues, with missing entries in the matrix corresponding to unobserved expression levels of the gene. We used a nonparametric method missForest[19] to impute the missing entries. Specifically, we first sorted the tissues in ascending order based on the number of missing samples and performed an initial imputation using the mean value of the observed samples. Then, for each tissue, we trained a random forest model to impute the missing values. For example, for the $k$th tissue (column), we trained a model using samples that have the non-missing entries in the $k$th column, with the $k$th column as the response and other columns as

predictors (with missing entries in other columns imputed). The missing entries of the $k$th column can be predicted using the trained model. We continue this iterative training process until convergence. For each gene, the final imputed matrix $\tilde{\mathbf{E}}$, which is of dimensions $N \times K$, is used for subsequent analyses.

### Identification of ct-eQTLs and ts-eQTLs

To identify ct-eQTLs, we perform Principal Component Analysis (PCA) on $\tilde{\mathbf{E}}$ of each gene. Then, we treat each of the top principal components (PCs) of $\tilde{\mathbf{E}}$ as the response variable and regress it against genotypes of cis-SNPs of the corresponding gene. We identify the set of ct-eQTLs minimizing:

$$EBIC_\lambda(\mathcal{S}) = -2\ell_N(\hat{\beta}_{\mathcal{S}}) + |\mathcal{S}|\log N + 2\lambda|\mathcal{S}|\log M, \ \mathcal{S} \supseteq \mathcal{S}_0, \quad (1)$$

where $\mathcal{S}$ is the selected set; $|\mathcal{S}|$ is the size of set $\mathcal{S}$; $\hat{\beta}_{\mathcal{S}}$ is the estimated effect of the selected variables; $\ell_N(\hat{\beta}_{\mathcal{S}})$ is the log-likelihood; $M$ is the number of cis-SNPs; $\lambda$ is the tuning parameter; and $\mathcal{S}_0$ is a pre-fixed set of cis-SNPs. When using the first PC as the response variable, we have $\mathcal{S}_0 = \varnothing$. As we proceed to the next PC, $\mathcal{S}_0$ encompasses the ct-eQTLs identified previously. This inclusion acts as a form of penalization, to avoid selecting an excessive number of ct-eQTLs. The number of PCs used to identify ct-eQTLs is determined by the magnitudes of eigenvalues of $\tilde{\mathbf{E}}$. Intuitively, an eigenvalue of 2 means that the corresponding PC explains about two variables' worth of the variability. We use 2.0 as a default cutoff, which leads to about 5 PCs on average in the GTEx studies. Supplementary Fig. 4a shows the number of the ct-eQTLs identified in GTEx cohorts as the number of PCs increases. We also evaluate the prediction performance across varying numbers of PCs (Supplementary Fig. 4b, c). In general, prediction accuracy improves as the number of PCs increases. However, beyond 4 PCs, the improvement levels off, suggesting that an average of 5 PCs strikes a good balance between prediction accuracy and computational efficiency.

Next, we identify ts-eQTLs. For each tissue, we regress its each gene's expression levels against genotypes of the gene's cis-SNPs, and select variables based on the EBIC criterion (Eq. (1)). The set $\mathcal{S}_0$ comprises all identified ct-eQTLs. We utilize a stepwise method for efficient computation, which follows the SODA procedure proposed in ref. [18] without considering interaction terms. We note that both imputed and observed samples are used in selecting ts-eQTLs. Despite the imputation is mostly capturing the tissue-shared part, it also involves interaction and nonlinear information that may carry tissue-specific component. We performed a replication study showing that including imputed samples in ts-eQTL identification has moderate benefits in prediction accuracy (Supplementary Table 3). The algorithm employs a forward step selecting predictors with significant overall effects, culminating in subsequent backward elimination steps to precisely refine the model. We adapt the algorithm by allowing a prior set of preselected terms, which ensures that terms selected in earlier steps are retained in the model. Effects of all selected terms are updated each time we introduce or eliminate a variable. The procedure is summarized in Algorithm 1.

**Algorithm 1** The algorithm for identifying ct-eQTLs and ts-eQTLs with EBIC criterion.

**Require:** A fixed term set $\mathcal{S}_0$, which can be $\varnothing$.

1: Forward procedure for selecting eQTLs. Let $\mathcal{M}_t$ denote the selected set of eQTLs at step $t$. Start with $\mathcal{M}_1 = \mathcal{S}_0$.

2: **while** not terminated **do**

3: **for** each $j \notin \mathcal{M}_t$ **do**

4: create a candidate set $\mathcal{M}_{t,j} = \mathcal{M}_t \cup \{j\}$, and evaluate its EBIC

5: **end for**

6: select predictor $j^*$ with the lowest EBIC: $j^* = \arg\min_j EBIC(\mathcal{M}_{t,j})$

7: **if** $EBIC(\mathcal{M}_{t,j^*}) < EBIC(\mathcal{M}_t)$ **then**

8: continue with $\mathcal{M}_{t+1} = \mathcal{M}_{t,j^*}$.

9: **else**

10: terminate and obtain set $\tilde{\mathcal{M}} = \mathcal{M}_t$.

11: **end if**

12: **end while**

13: Backward procedure for eliminating unimportant terms. Let $\mathcal{S}_t$ denote the selected set at the step $t$ of the backward stage. Start with $\mathcal{S}_1 = \tilde{\mathcal{M}}$.

14: **while** not terminated **do**

15: **for** each $j \in \mathcal{S}_t \backslash \mathcal{S}_0$ **do**

16: create a candidate set $\mathcal{S}_{t,j} = \mathcal{S}_t \backslash \{j\}$, and evaluate its EBIC

17: **end for**

18: find term $j$ with the lowest EBIC: $j^* = \arg\min_j \text{EBIC}(\mathcal{S}_{t,j})$

19: **if** $\text{EBIC}(\mathcal{S}_{t,j^*}) < \text{EBIC}(\mathcal{S}_t)$ **then**

20: remove term $j^*$

21: **else**

22: terminate and retain set $\tilde{S} = S_t$.

23: **end if**

24: **end while**

25: Enumerate all possible combinations of non-fixed terms and find the subset that reaches the smallest EBIC

$$\hat{\mathcal{A}} = \arg\min_{\mathcal{A} \subset \tilde{S} \backslash \mathcal{S}_0} \text{EBIC}(\mathcal{A} \cup \mathcal{S}_0)$$

With the selected ct-eQTLs and ts-eQTLs, we perform weighted least squares to estimate their effect sizes. Specifically, for a tissue with an observed sample size $N_{obs}$ (the number of non-missing entries), and an imputed sample size $N_{imp}$ (the number of missing entries), we assign a weight of $\min(1, N_{obs}/N_{imp})$ to the imputed samples, and a weight of 1 to the observed samples. This weighting scheme prioritizes the impact of the observed data, especially when the imputed data volume surpasses the observed data, so as to retain tissue-specific signals in estimating eQTL effects. We show that it improves the prediction accuracy compared with estimating effect sizes using only observed samples, in both GTEx studies ($P < 0.05$ for 46 out of 47 tissues, one-sided paired Wilcoxon test for prediction $R^2$) and the replication study (see Supplementary Table 3).

## Gene-trait association studies

For each trait, we compute gene-level summary statistics based on the training weights for the eQTLs derived by MTWAS. We denote the sample size as $N$. For each gene on each tissue, we assume a linear model between a complex phenotype (**Y**) and the gene expression (**E**):

$$\mathbf{Y} = \mathbf{E}\gamma + \eta, \tag{2}$$

where **Y** is an $N \times 1$ phenotype vector, **E** is an $N \times 1$ expression vector, and $\gamma$ is the gene-level effect size. We assume both **Y** and **E** have been standardized, and the error term $\eta$ follows a normal distribution with mean 0. The MTWAS $Z$-score vector is:

$$Z = \frac{\hat{\gamma}}{\text{se}(\hat{\gamma})}. \tag{3}$$

If individual-level genotype data are accessible, we could directly estimate gene expression with $\hat{\mathbf{E}} = \sum_{j \in \mathcal{A}} \hat{\beta}_j X_j$, where $X_j$ is the genotype of the $j$th SNP; $\hat{\beta}_j$ is the estimated effects on gene expression of the $j$th SNP; and $\mathcal{A}$ is the set of identified eQTLs for the gene. Then we regress **Y** against $\hat{\mathbf{E}}$ to derive the association between the trait and the gene expression. If the individual-level genotype data are not available, we could derive MTWAS test statistics with GWAS summary statistics. Specifically, the MTWAS $Z$-score can be approximated with[2,40]

$$Z = \frac{\hat{\gamma}}{\text{se}(\hat{\gamma})} \approx \sum_{j \in \mathcal{A}} \hat{\beta}_j \frac{\hat{\sigma}_j}{\hat{\sigma}} z_j, \tag{4}$$

where $z_j$ is the $z$-score for the $j$th SNP in GWAS summary statistics; $\hat{\sigma}_j^2$ is the sample variance of SNP $j$; $\hat{\sigma}^2$ is the sample variance of the gene expression. We remove the major histocompatibility complex region (6p21.3; GRCh38 coordinates 6: 28,510,120–33,480,577) in the TWAS analysis of UKBB phenotypes. We apply the Bonferroni correction to account for multiple testing. The independent TWAS genes are extracted by calculating squared Pearson correlation ($r^2$) between the predicted expressions of all gene pairs within each tissue. For any gene pair with $r^2 > 0.5$, we only keep the gene with the lower TWAS $P$ value[41].

## Compared methods

**PrediXcan.** PrediXcan is a TWAS method testing the molecular mechanisms through which genetic variation affects phenotype[2]. For each gene on each target tissue, PrediXcan trains a prediction model with elastic net, using the accessible genome variation and gene expression levels. Barbeira et al.[3] extend the PrediXcan to S-PrediXcan, which can be applied when only GWAS summary statistics are available. The PrediXcan software is available at https://github.com/hakyimlab/PrediXcan/tree/master/Software.

**TIGAR.** TIGAR is an improved Bayesian tool for transcriptomic data prediction. Specifically, they adopt a nonparametric Bayesian approach by assuming a Dirichlet process prior for the distribution of the effect-size variance[9,10]. The method can flexibly model the genetic architecture of gene expression levels, and is more general than elastic net and other Bayesian methods with a specific prior. The TIGAR software is available at https://github.com/yanglab-emory/TIGAR. We implemented TIGAR with the default settings.

**UTMOST.** UTMOST is a TWAS method that trains a cross-tissue expression prediction model by using the genotype information and matched expression data from multiple tissues[13]. Specifically, the cross-tissue expression prediction is formulated as a penalized multivariate regression problem, and effect sizes are estimated by minimizing a squared loss function with a lasso penalty on columns (within-tissue effects) and a group-lasso penalty on rows (cross-tissue effects). The UTMOST software is available at https://github.com/Joker-Jerome/UTMOST.

## Data preprocessing

We followed the quality control and sample exclusion process provided by the GTEx portal for the genotype and gene expression datasets[6]. SNPs with minor allele frequency (MAF) <0.05 or with strand-ambiguities were removed. We examine cis-SNPs located within a genomic range of ±1 base pair from the gene and overlapped with HapMap Project Phase 3 SNPs. For the GTEx dataset, gene expression data of 17,566 genes were normalized through rank-based inverse normal transformation, and were further adjusted for sex, 3 genotyping PCs, and top 15 PEER factors (to quantify batch effects and experimental confounders)[2,42]. We trained models based on 47 tissues with European ancestry sample sizes larger than 100. The GEUVADIS dataset was used for external validation, and we focused on the 373 individuals of European ancestry. For the DICE dataset, we utilized 2 genotyping PCs as covariates as suggested in ref. 21. A total of 19,020 genes were evaluated. For the OneK1K dataset, in line with the guidelines provided by Yazar et al.[22], we excluded genes expressed in fewer than 10% of the cohort for each cell type. We focused on genes that were retained in more than one cell type after filtering. The expression values were log-transformed ($\log(x+1)$). The gene expression was subsequently adjusted for sex, age, 6 genotyping PCs, and 2 PEER factors.

## Model training and evaluation

For the GTEx, DICE, and OneK1K studies, we used the fivefold CV for performance evaluation. Specifically, we randomly divided data into five subsets. For each fold, we performed imputation on four of these subsets designated as training data, and the remaining subset was used for testing, which were unimputed. The performance of predicting gene expression levels from genotypes was evaluated with the prediction $R^2$ and the number of predictable genes. We considered two criteria for identifying predictable genes, including prediction $R^2 > 0.01$, which is a common standard widely used in TWAS analysis such as PrediXcan[2]. We also considered a stringent criterion, where we performed an $F$ test with degrees of freedom 1 and $N - 2$ to assess the significance of the predictability, where $N$ is the sample size. The $P$ values were then adjusted with the Benjamini–Hochberg (BH) procedure to control the FDR at a 0.05 level[43].

In addition to the internal CV analysis, we conducted an external validation study with the GEUVADIS dataset, which consists of LCLs of 373 individuals of European ancestry. We trained the model on the EBV transformed lymphocytes from the GTEx dataset ($N = 116$).

## Pathway enrichment analysis

We tested the enrichment of ct-genes and ts-genes in the KEGG pathway using the R package clusterProfiler[44,45]. We employed the BH procedure to control the FDR at a 0.05 level[43].

## Reporting summary

Further information on research design is available in the Nature Portfolio Reporting Summary linked to this article.

## Data availability

The MTWAS summary statistics for 84 UKBB phenotypes are provided in Supplementary Data 2. The eQTL weights generated by MTWAS, utilizing the GTEx v8, DICE, OneK1K datasets are available at Zenodo[46] (https://doi.org/10.5281/zenodo.11647460). The genotype and gene expression data of GTEx v8 project were downloaded from the database of Genotypes and Phenotypes (dbGaP) under accession number phs000424.v8.p2. The genotype and gene expression data of GEU-VADIS LCLs were downloaded from the EBI ArrayExpress database under accession code E-GEUV-1. The DICE project provides anonymized gene expression data for public access at https://dice-database.org. The genotype data can be accessed from the dbGaP under accession number phs001703.v4.p1. The OneK1K single-cell gene expression and genotype data are available via Gene Expression Omnibus under accession number GSE196830. GWAS summary statistics from the UKBB were downloaded from the repository at http://www.nealelab.is/uk-biobank. The effective sample sizes for the binary UKBB phenotypes were calculated by $\frac{4n_{case}n_{control}}{n_{case}+n_{control}}$. The LD matrix was estimated with UKBB European ancestry samples, which can be downloaded from https://pan.ukbb.broadinstitute.org[47]. We partitioned the genome into 1703 independent blocks using LDetect[48] at https://bitbucket.org/nygcresearch/ldetect-data/src/master/, based on the 1000G reference panel with European ancestry. Source data are provided with this paper.

## Code availability

An R package implementing MTWAS is available on the GitHub repository (https://github.com/szcf-weiya/MTWAS) and Zenodo[46] (https://doi.org/10.5281/zenodo.11647460).

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

## Acknowledgements

The Genotype-Tissue Expression (GTEx) Project was supported by the Common Fund of the Office of the Director of the National Institutes of Health, and by NCI, NHGRI, NHLBI, NIDA, NIMH, and NINDS. This research has been conducted using the publicly available UK Biobank GWAS summary statistics released by Neale's lab. This work was supported by the National Key R&D Program of China (2021YFF1200905) to L.H. and by the National Institute of General Medical Sciences Grant R01 GM152814-01 to J.S.L.

## Author contributions

S.S., L.H., and J.S.L. conceived the study. S.S. developed the statistical methods and performed simulations and real data analyses with guidance from L.H. and J.S.L. S.S. and L.J.W. developed the MTWAS software. S.S. and L.H. applied the data. All authors contributed to this work in data analysis, result interpretation, and manuscript writing. All authors have reviewed and approved the final version of the manuscript.

## Competing interests

The authors declare no competing interests.
