## [Peer Review File · Nature Communications]

REVIEWER COMMENTS

Reviewer #1 (Remarks to the Author):

The authors have proposed a statistical method for multi-tissue transcriptome prediction, in which they first augmented unobserved gene expressions a non-parametric method and then selected cross-tissue and tissue specific eQTLs for prediction. They have shown that MTWAS improves imputation accuracy in GTEx tissues compared to existing methods. I think the proposed approach is interesting to firstly augment the transcriptome and then build the prediction model. However, I have some concerns,

1. First, the definition of cross-tissue eQTLs and tissue specific eQTLs is not so clear. Do cross-tissue eQTLs regulate gene expression more than one-tissue? The authors first performed PCA and then use top PCs to select cross-tissue eQTLs. What's the rational that PCs corresponding to the cross-tissue eQTLs? Usually PCA in gene expression analysis identifies batch effect. What do the PCs mean here?

2. The authors stated they used SODA [Li and Liu 2019] for variable selection, but they did not consider interaction terms. They fixed main effect terms and exhaustively search for the variable set to achieve the smallest EBIC. I'm confused that if the main effect terms were fixed and interaction terms were not considered, what is variable selection approach optimizing?

Reviewer #2 (Remarks to the Author):

The authors proposed a new MTWAS method to leverage transcriptomic data of multiple tissues to train tissue-specific gene expression imputation models, which classifies eQTLs into cross-tissue and tissue-specific via a step-wise procedure based on the extended Bayesian information criterion. Real applications with the GTEx V8 reference data and UKBB GWAS summary data showed preferred performance by MTWAS, compared to the UTMOST and PrediXcan methods. The methods and results are appealing. However, a fairer comparison with existing methods and details showing the advantages of the final TWAS association study results are needed. Please see my comments as follows:

1. As the authors stated in the manuscript that there are three possible aspects that could lead to the improved performance by MTWAS, I would suggest the comparison with UTMOST using the imputed gene expression data and TIGAR/DPR weights (instead of PrediXcan) with GTEx V8 (PMID: 35047855) for a fairer comparison. As shown by the TIGAR paper, the non-parametric Bayesian DPR model can train gene expression imputation models with higher CV R2 and obtain valid gene expression imputation models for more genes. The Bayesian DPR model might be more comparable to the extended Bayesian information criterion used by MTWAS, than the elastic-net model. Since both MTWAS and UTMOST would use the gene expression data of all available tissues, using the same imputed gene expression data to estimate eQTL weights should be done in the comparison.

2. The authors should state the computation cost of the MTWAS method for the GTEx V8 data, and how it would be compared to UTMOST.

3. Although the authors discussed the enrichment analysis results and CADD/functional scores for the cross-tissue and tissue-specific eQTLs, the authors might also discuss how the cross-tissue and tissue-specific eQTL would affect the association results with the UKBB. Comparing the association study results by MTWAS to MTWAS-tissue would probably show more information.

4. The authors might show Venn diagrams about how the significant genes identified by MTWAS, UTMOST, and TIGAR/DPR would overlap? And discuss the association study results for a few example traits in more detail. For now, the advantages of MTWAS about having a higher average CV R2 for the gene expression imputation models, and more numbers of valid gene expression imputation models are shown in real data. However, the advantages of MTWAS with the final association study results are unclear. Also, identifying a larger number of total significant genes might not necessarily mean more independent significant genes. Significant genes with shared eQTL are likely not independent significant ones.

5. The eQTL weights generated by MTWAS with GTEx V8, the source code of MTWAS, and TWAS summary data with UKBB data should be made publicly available.

Reviewer #3 (Remarks to the Author):

In this manuscript, Song and colleagues proposed a novel method to build prediction models of gene expressions in the context of multi-tissue datasets like GTEx. Via the analysis on 47 tissues in

GTEX v8, the authors show that the proposed models outperform existing widely used methods: elastic nets (in PrediXcan) and UTMOST (a cross-tissue method).

As a method paper, it presented the method and evaluated the performance nicely but the problem that the method is trying to tackle is slightly outdated in the sense that the field has moved towards single-cell data so a bulk data-based approach is fundamentally less interesting to people. To make this work more impactful, it'd be great if the authors could run and share their models on other multi-tissue/multi-cell type datasets such as PsychENCODE or DICE. Besides, showcasing the ability to process single-cell data would be of great significance.

Major comments:

1. For a tissue-specific expression, how well the imputation will perform? As it leverages info from other tissues to make predictions, would it only capture the tissue-shared component of the expression?
2. In Figure 3 Uterus panel, maybe I'm not understanding it correctly but it seems strange to me that Uterus has more imputable genes than Muscle Skeletal even in PrediXcan. It seems that something is off. Similarly, it is unexpected to see small-sample-size tissues having a lot more imputable genes than tissue with a bigger sample size in Figure S2.
3. For the RNF186 association in UC, the reported discovery is in the ovary which does not sound related to UC. Did you observe consistent results (sign or nominal significance) in more relevant tissue?
4. Considering EBIC as a variable selection approach, how does it perform compared to a simpler approach like forward selection or a fine-mapping approach like SuSiE?

Minor comments:

1. At line 70 "Then, ...", the first part of this sentence ("we impute the missing entries ..") is confusing to me since the previous sentence has mentioned the imputation step so it sounds duplicated.
2. At line 77 "We use ...", this "2" sounds like a GTEX-specific parameter? As in other data sets, we may not have ~50 tissues to begin with so that the number should be changed? It might be worth to make this clearer and more generic rather than presented as a magic number.
3. At line 81, it is unclear what "tissue-wise" means here. From my understanding, this step is done one tissue at a time (Step 3 in Figure 1) so this is not tissue-wise.
4. When using R^2 for performance evaluation, it is better to consider the sign, i.e. signed R^2 .
5. In identifying ts-eQTL, what's the motivation for fixing the eQTLs identified in previous PCs for the analysis of the next PCs? It's not clear to me why the next PC needs to have the eQTLs from all previous PCs as eQTLs.

We thank the editors for organizing this quick and thorough review of our paper. We are very grateful to the referees for their dedication, thoughtful comments, and constructive suggestions. In the revised manuscript, we have attended to all the comments and revised the manuscript accordingly. We have highlighted the changes in our manuscript. Point-by-point responses to the comments are provided below.

Responses to the comments of Reviewer #1

Comments to the Author

The authors have proposed a statistical method for multi-tissue transcriptome prediction, in which they first augmented unobserved gene expressions a non-parametric method and then selected cross-tissue and tissue specific eQTLs for prediction. They have shown that MTWAS improves imputation accuracy in GTEx tissues compared to existing methods. I think the proposed approach is interesting to firstly augment the transcriptome and then build the prediction model. However, I have some concerns,

Thank you very much for your positive and thoughtful comments. As explained below, we have carefully thought through all your comments and addressed each of them. Your comments greatly helped us in this revision to better explain intuitions and to improve the clarity of the manuscript.

Major comments:

1. First, the definition of cross-tissue eQTLs and tissue specific eQTLs is not so clear. Do cross-tissue eQTLs regulate gene expression more than one-tissue? The authors first performed PCA and then use top PCs to select cross-tissue eQTLs. What's the rationale that PCs corresponding to the cross-tissue eQTLs? Usually PCA in gene expression analysis identifies batch effect. What do the PCs mean here?

Thanks for your comments and questions. We define cross-tissue eQTLs as genetic variations that are associated with gene expression in multiple tissues. Tissue-specific eQTLs are defined as the genetic variations associated with gene expression within a particular tissue, after accounting for the effects of cross-tissue eQTLs. In practice, we identify cross-tissue eQTLs by regressing top principal components of the sample-by-tissue matrix against the genotypes of cis-SNPs for each gene. Intuitively, PCA is used to extract the shared information across multiple tissues. In other words, top principal components (PCs) reflect most common effects that may include both batch effects (which may be regarded as “cross-tissue” effects as well), if there is any, and cross-tissue information.

For a better illustration, we plot in Figure 1 below a 47×50 matrix, corresponding to PCA loadings of the 47 GTEx tissues onto the first 5 PCs of the 10 most predictable genes (i.e., those with the highest prediction R^2 averaged over the 47 tissues). Note that, for each gene, the loading of tissue i onto PC_k is just the covariance between the sample vector of tissue i and PC_k of that

gene. For each gene, the loadings of the first PC are evenly distributed across 47 tissues, hence we can interpret the first PC as the average expression level concerning all tissues. This can also be observed in predictable genes of specific tissues. We provide two examples in Figures 2 and 3 below for predictable genes in the whole blood and brain cortex. Similarly, the first PC characterizes the overall cross-tissue expression patterns, and explains 89.6% (SD=3.7%) and 86.9% (SD=7.1%) of the total variance, respectively. The other PCs more likely reflect differences between subsets of tissues (e.g., brain-related tissues versus the others).

We clarify that the PCA used here is different from that used for identifying batch effects. For identifying the batch effects, PCA is applied to the sample-by-gene matrix within each tissue separately^{[1][2]}. In contrast, our cross-tissue PCA is performed on the sample-by-tissue matrix for each gene to extract the cross-tissue information. Furthermore, we have already corrected the batch effects of the data in the preprocessing procedure by applying a probabilistic estimation of expression residuals (PEER) suggested in the GTEx protocol^[3].

We have added the definition of cross-tissue eQTLs and tissue-specific eQTLs, and the rationale of PCA in Lines 74-85 on Page 3. We have also provided some clarifications on the differences between our cross-tissue PCA and the PCA used in correcting batch effects in Lines 297-300 on Page 14.

Figure 1. The heatmap of PCA loadings of the first 5 PCs in the 10 most predictable genes across 47 GTEx tissues.

Figure 2. The heatmap of PCA loadings of the first 5 PCs in the 5 most predictable genes in the GTEx whole blood tissue.

Figure 3. The heatmap of PCA loadings of the first 5 PCs in the 5 most predictable genes in the GTEx brain cortex tissue.

2. The authors stated they used SODA [Li and Liu 2019] for variable selection, but they did not consider interaction terms. They fixed main effect terms and exhaustively search for the variable set to achieve the smallest EBIC. I'm confused that if the main effect terms were fixed and interaction terms were not considered, what is variable selection approach optimizing?

Thanks for the question, which gives us an opportunity to clarify our previous writing. We are sorry for not explaining our procedure well. Briefly speaking, our variable selection approach can be regarded as a two-stage-procedure, and each stage aims at finding a subset:

$$\hat{A} = \operatorname{argmin}_{A \subset S \setminus S_0} EBIC(A \cup S_0),$$

where S is the set of cis-SNPs. Notation S_0 represents the set of fixed terms, which has different meanings in each stage (The detailed description of the algorithm is provided on Page 16).

The first stage is to identify cross-tissue eQTLs. Assuming we consider the first 5 PCs of a cross-tissue expression matrix. When considering the first PC as the response variable, we set $S_0 = \emptyset$. For the second PC, S_0 is updated to the variables (eQTLs) identified in association with the first PC; and for the third PC, S_0 is further expanded to encompass variables identified for the first two PCs, etc. Therefore, the variables selected for the $(k+1)$ -th PC contains those selected for the k -th PC prefixed in their respective S_0 . The second stage identifies tissue-specific eQTLs, in which S_0 represents the cross-tissue eQTL set we found in the first stage.

We leverage the idea of SODA in each stage with forward steps to select predictors with significant overall effects, and backward elimination steps for further pinning down the exact model^[4]. Our minor twist is to add a “prior” set S_0 of pre-selected main effects. Nevertheless, we do not really fix any main effects in advance, a “fixed set” only appears in our step-wise procedure as a way to keep those terms selected in previous steps. The effects of all selected terms are updated each time we introduce or eliminate a variable. We have added some clarifications in Lines 350-353 on Page 15.

References:

- [1] Flutre, T., Wen, X., Pritchard, J., & Stephens, M. (2013). A statistical framework for joint eQTL analysis in multiple tissues. *PLoS Genetics*, 9(5), e1003486.
- [2] Pickrell, J. K., Marioni, J. C., Pai, A. A., Degner, J. F., Engelhardt, B. E., Nkadori, E., ... & Pritchard, J. K. (2010). Understanding mechanisms underlying human gene expression variation with RNA sequencing. *Nature*, 464(7289), 768-772.
- [3] Stegle, O., Parts, L., Piipari, M., Winn, J., & Durbin, R. (2012). Using probabilistic estimation of expression residuals (PEER) to obtain increased power and interpretability of gene expression analyses. *Nature Protocols*, 7(3), 500-507.
- [4] Li, Y., & Liu, J. S. (2019). Robust variable and interaction selection for logistic regression and general index models. *Journal of the American Statistical Association*, 114(525), 271-286.

Responses to the comments of Reviewer #2

Comments to the Author

The authors proposed a new MTWAS method to leverage transcriptomic data of multiple tissues to train tissue-specific gene expression imputation models, which classifies eQTLs into cross-tissue and tissue-specific via a step-wise procedure based on the extended Bayesian information criterion. Real applications with the GTEx V8 reference data and UKBB GWAS summary data showed preferred performance by MTWAS, compared to the UTMOST and PrediXcan methods. The methods and results are appealing. However, a fairer comparison with existing methods and details showing the advantages of the final TWAS association study results are needed. Please see my comments as follows:

Thank you very much for your positive and constructive comments. We have provided the point-to-point responses below.

Major comments:

1. As the authors stated in the manuscript that there are three possible aspects that could lead to the improved performance by MTWAS, I would suggest the comparison with UTMOST using the imputed gene expression data and TIGAR/DPR weights (instead of PrediXcan) with GTEx V8 (PMID: 35047855) for a fairer comparison. As shown by the TIGAR paper, the non-parametric Bayesian DPR model can train gene expression imputation models with higher CV R2 and obtain valid gene expression imputation models for more genes. The Bayesian DPR model might be more comparable to the extended Bayesian information criterion used by MTWAS, than the elastic-net model. Since both MTWAS and UTMOST would use the gene expression data of all available tissues, using the same imputed gene expression data to estimate eQTL weights should be done in the comparison.

Thanks for your suggestion. Based on your suggestion, we have added the comparison of three types of weights: (1) UTMOST trained with imputed data (UTMOST-imp); (2) TIGAR/DPR weights with the original data (TIGAR^{[1][2]}); and (3) TIGAR/DPR weights with imputed data (TIGAR-imp). The results are provided in Figures 1 and 2 below. Across all 47 tissues, MTWAS still shows the best performance among all the tested methods. Indeed, we see the superiority of TIGAR over PrediXcan especially across tissues with larger sample sizes. Nevertheless, methods integrating cross-tissue information shows better performance in tissues especially with smaller sample sizes.

The new results have demonstrated our original points. First, our nonparametric imputation of the expression matrices utilizes the cross-tissue information and increases the effective sample sizes of the training dataset. The improvement can be shown from the comparison between PrediXcan and PrediXcan-imp, UTMOST and UTMOST-imp, and TIGAR and TIGAR-imp. For all 47 tissues, the methods with our imputed data perform better than the original ones. Second, from the comparison between MTWAS, PrediXcan-imp, UTMOST-imp, and TIGAR-imp, we could see the superiority

of EBIC criterion under high-dimensional settings. We have added the results in Lines 110-129 on Pages 5-6.

We have updated Figure 2 in our manuscript (Figure 1 below), and added a Supplementary Figure 5 (Figure 2 below) to compare three types of weights in all 47 GTEx tissues.

Figure 1. Improvements of prediction R^2 over PrediXcan, of MTWAS, UTMOST, and TIGAR (DPR), evaluated on 47 tissues of GTEx datasets. The tissues are arranged in descending order based on their respective sample sizes, from the highest (muscle skeletal) to the lowest (uterus). The prediction R^2 is based on 5-fold CV.

Figure 2. Improvements of prediction R^2 over PrediXcan, of MTWAS, MTWAS-tissue, PrediXcan-imp, UTMOST, UTMOST-imp, TIGAR, and TIGAR-imp, evaluated in the GTEx datasets. The dashed line marks the performance of MTWAS. The tissues are arranged in descending order based on their sample sizes, from the highest (muscle skeletal) to the lowest (uterus). The prediction R^2 is based on 5-fold CV.

2. The authors should state the computation cost of the MTWAS method for the GTEx V8 data, and how it would be compared to UTMOST.

Thanks for your suggestion. Table 1 below shows the computational time of MTWAS and UTMOST for the 47 tissues in the GTEx data based on chromosome 1 (1,911 genes) and chromosome 22 (401 genes). For MTWAS, we reported the CPU time for (i) imputing the missing entries of the expression matrices; and (ii) identifying eQTLs and estimating the eQTL weights, while UTMOST only involves the second process. Taking chromosome 1 (1,911 genes) as an example, it takes MTWAS about 18 minutes to impute missing entries. The eQTL identification and weights estimation process take about 45 minutes for MTWAS, which is about half the time required by UTMOST (87 minutes). We have added a section on computational efficiency on Page 13.

Table 1. Runtime (minutes) comparison for MTWAS and UTMOST. Both methods were applied to 47 tissues in the GTEx dataset on chromosome 1 and chromosome 22. MTWAS first imputes the missing entries in expression matrices. Both methods include identifying eQTLs and estimating eQTL weights on gene expression. The computation was performed with an Intel Xeon processor with 2.90GHz and 128 cores.

	MTWAS		UTMOST
	Imputation	Model Training	Model Training
chromosome 1 (1,911 genes)	18	45	87
chromosome 22 (401 genes)	4	9	18

3. Although the authors discussed the enrichment analysis results and CADD/functional scores for the cross-tissue and tissue-specific eQTLs, the authors might also discuss how the cross-tissue and tissue-specific eQTL would affect the association results with the UKBB. Comparing the association study results by MTWAS to MTWAS-tissue would probably show more information.

Thank you for the suggestion. MTWAS-tissue differs from MTWAS in that it does not partition the ct-eQTLs and ts-eQTLs. Instead, it directly performs variable selection with all cis-SNPs on each tissue. As is shown in Figure 2 on Pages 7-9 of the response letter, MTWAS achieves higher prediction accuracy than MTWAS-tissue by integrating cross-tissue eQTLs, which also results in more predictable genes. We further compare the association study results by MTWAS and MTWAS-tissue in 84 UKBB self-reported cancer and non-cancer illness phenotypes with effective sample sizes larger than 5,000. On average, MTWAS identifies 203 significant gene-tissue pairs, whereas MTWAS-tissue only identifies 170 significant gene-tissue pairs. Taking the heart attack/myocardial infarction in the UKBB as an example, MTWAS identified 154 significant gene-tissue pairs after Bonferroni correction, while MTWAS-tissue only identified 120 gene-tissue pairs.

We note that the expression level of a gene can be affected by cross-tissue eQTLs and tissue-specific eQTLs simultaneously, so we consider two subsets of the predictable genes that associated with a specific phenotype, according to whether the genes are regulated only by ct-eQTLs or ts-eQTLs, which we refer to as cross-tissue genes (ct-genes) and tissue-specific (ts-genes), correspondingly. We found that the ct-genes associated with heart attack/myocardial infarction were enriched in metabolism and lysosome pathways, which is consistent in our pathway analysis on the predictable ct-genes in GTEx studies. As for the ts-genes, more tissue-specific pathways were observed (Figure 3 below). We have added the results in Lines 225-231 on Page 11.

Figure 3. KEGG pathway enrichment analysis of cross-tissue genes and tissue-specific genes that associated with heart attack/myocardial infarction. The significance of enrichment is depicted through the color of the dots, and the gene ratio is indicated by the horizontal distance between the dots and the y-axis. The sizes of the dots correspond to the gene counts.

4. The authors might show Venn diagrams about how the significant genes identified by MTWAS, UTMOST, and TIGAR/DPR would overlap? And discuss the association study results for a few example traits in more detail. For now, the advantages of MTWAS about having a higher average CV R2 for the gene expression imputation models, and more numbers of valid gene expression

imputation models are shown in real data. However, the advantages of MTWAS with the final association study results are unclear. Also, identifying a larger number of total significant genes might not necessarily mean more independent significant genes. Significant genes with shared eQTL are likely not independent significant ones.

Thanks for your insightful comments and nice suggestions. In our previous version of the manuscript, we applied MTWAS, UTMOST, and PrediXcan to 84 UKBB self-reported cancer and non-cancer illness phenotypes with effective sample sizes larger than 5,000, and MTWAS identified significant genes in more phenotypes compared with UTMOST and PrediXcan (see Lines 209-223 on Pages 10-11 of the revised manuscript).

Taking your suggestion, we now elaborate more on genes that are significantly associated with heart attack/myocardial infarction (MI) in the UKBB (Figure 4 below). We further identified independent genes by calculating squared Pearson correlation (r^2) between the predicted expression for each pair of genes for each tissue^[3]. For a pair of genes with the predicted expression $r^2 > 0.5$, we only kept the one with the smaller TWAS p-value. MTWAS identified 30 independent genes (Table 1 below) and 114 gene-tissue pairs that are associated with MI, among which 7 were not identified by UTMOST, PrediXcan, or TIGAR. We highlight some genes that are identified only by MTWAS. For example, MTWAS identified the *BRCA-1* associated protein gene, *BRAP* ($p=1.87 \times 10^{-10}$ in artery aorta and 5.39×10^{-8} in artery coronary), is crucial for both cardiac development and myocardial function. The absence of *BRAP*, as observed in knockout models, resulted in the arrest of the cell cycle, diminished proliferation of cardiomyocytes, and early onset of heart failure^[4]. Conversely, the introduction of transgenic *BRAP* overexpression led to an augmentation in myocardial mass and increased cell cycle activity specifically in neonatal cardiomyocytes. Given the critical role of the aorta in systemic circulation and the coronary arteries in supplying oxygen-rich blood to the myocardium, the dysregulation of *BRAP* expression in arterial tissues could disrupt the delicate equilibrium of cardiomyocyte growth and function, potentially leading to the development and progression of heart failure. Another example is that prohibitin (*PHB*) ($p=4.66 \times 10^{-8}$ in adipose subcutaneous) has emerged as a potential therapeutic target for diabetic cardiomyopathy (DCM). In a type 2 diabetic rat model, *PHB* overexpression alleviated insulin resistance, left ventricular dysfunction, and fibrosis, suggesting its promise in treating human DCM^[5]. In addition, *FOSB* (significant in multiple tissues) has been identified as a potential diagnostic biomarker and therapeutic target for heart failure in a recent study^[6]. We also notice that *ERP29* ($p=7.12 \times 10^{-9}$ in artery tibial), involved in Connexin43 (Cx43) hemichannel assembly, plays a crucial role in Cx43 stability. Dysregulation of Cx43 is linked to myocardial diseases, and *ERP29*'s function suggests a potential connection to these conditions, providing insights into connexin-related myocardial disorders^[7]. We have added the results in Lines 225-250 on Pages 11-12.

Figure 4. Venn diagram of genes significantly associated with heart attack/myocardial infarction in the UKBB, identified by MTWAS, UTMOST, PrediXcan, and TIGAR.

Table 1. Independent TWAS risk genes of heart attack/myocardial infarction in the UKBB, identified by MTWAS. The most significant tissue and p-value denote the tissue with the highest significance level for each gene. Literature as evidence indicating the associations between the identified gene and MI is provided.

Gene	chr	Most significant tissue	p-value	# tissues detected	Predi Xcan	UTM OST	TIGA R	Evidence
AIDA	1	Brain Cerebellar	6.18E-11	2	✓	✓		[8]
CELSR2	1	Muscle Skeletal	2.18E-09	4	✓	✓		[9]
FAM177B	1	10 GTEx tissues*	9.10E-11	10	✓	✓	✓	[10]
MIA3	1	Cells Cultured fibroblasts	1.39E-13	37	✓	✓	✓	[11]
PSMA5	1	Liver, Nerve Tibial	4.77E-08	2	✓			
PSRC1	1	Esophagus Muscularis	4.56E-10	7	✓	✓	✓	[9]
SORT1	1	Heart Left Ventricle	2.71E-09	10	✓	✓		[12]
LPA	6	Liver	5.23E-10	1	✓	✓		[13] [14]
PHACTR1	6	Artery Tibial	1.36E-13	2		✓		[15]
CDKN2B	9	Brain Anterior, Small Intestine Terminal Ileum	4.11E-29	9	✓	✓	✓	[16]
IFNA10	9	Artery Aorta	2.42E-16	1		✓	✓	
IFNA13	9	Brain Nucleus	5.60E-09	1			✓	
IFNA14	9	Esophagus Gastroesophageal Junction	2.30E-19	1		✓	✓	
IFNA16	9	Skin Sun Exposed Lower leg	2.30E-19	3		✓	✓	
IFNA17	9	Brain Amygdala	1.16E-09	1	✓			
IFNA2	9	Brain Cerebellum	2.30E-19	3			✓	
IFNA5	9	Brain Putamen	8.80E-20	2	✓	✓		
IFNA6	9	Brain Hippocampus	1.65E-11	3			✓	
IFNA7	9	Cells Cultured fibroblasts	8.58E-14	1			✓	

IFNA8	9	Heart Left Ventricle	8.80E-20	1		√	√	
IFNB1	9	Esophagus Mucosa	3.06E-11	1	√			
IFNW1	9	Brain Putamen	1.54E-11	1		√	√	
BRAP	12	Artery Aorta	1.87E-10	2				[17] [18] [19]
ERP29	12	Artery Tibial	7.12E-09	1				[20]
PPP1CC	12	Adrenal Gland	7.12E-09	1				
FES	15	Cells Cultured fibroblasts	1.35E-08	1	√	√	√	[21]
PHB	17	Adipose Subcutaneous	4.66E-08	1				[22]
APOC1	19	Brain Nucleus	6.97E-09	1				[23]
FOSB	19	Brain Cortex	4.04E-08	3				[24]
MYPOP	19	Cells Cultured fibroblasts	4.04E-08	1				

* Brain Amygdala, Brain Nucleus, Cells Cultured fibroblasts, Cells EBV transformed, lymphocytes, Esophagus Gastroesophageal Junction, Liver, Lung, Minor Salivary Gland, Prostate, Whole Blood.

5. The eQTL weights generated by MTWAS with GTEx V8, the source code of MTWAS, and TWAS summary data with UKBB data should be made publicly available.

Thank you for your suggestion. We have provided the eQTL weights generated by MTWAS with GTEx v8, the source code of MTWAS in our GitHub repository (<https://github.com/szcfweiya/MTWAS>). The TWAS summary data with UKBB data are available in Supplementary Table 2. We have provided the links in the “Data availability” and “Code availability” sections on Page 19.

References:

- [1] Nagpal, S., Meng, X., Epstein, M. P., Tsoi, L. C., Patrick, M., Gibson, G., ... & Yang, J. (2019). TIGAR: an improved Bayesian tool for transcriptomic data imputation enhances gene mapping of complex traits. *The American Journal of Human Genetics*, 105(2), 258-266.
- [2] Parrish, R. L., Gibson, G. C., Epstein, M. P., & Yang, J. (2022). TIGAR-V2: Efficient TWAS tool with nonparametric Bayesian eQTL weights of 49 tissue types from GTEx V8. *Human Genetics and Genomics Advances*, 3(1).
- [3] Dai, Q., Zhou, G., Zhao, H., Vösa, U., Franke, L., Battle, A., ... & Yang, J. (2023). OTTERS: a powerful TWAS framework leveraging summary-level reference data. *Nature Communications*, 14(1), 1271.
- [4] Grebe, C., Schott, P., Unsöld, B., Männer, J., Didie, M., Kurz, K., ... & Seidler, T. (2012). Brca1-associated Protein 2 (BRAP) is Essential for Embryonic Heart Development and for Neonatal Cardiomyocyte Proliferation.
- [5] Dong, W. Q., Chao, M., Lu, Q. H., Chai, W. L., Zhang, W., Chen, X. Y., ... & Zhang, M. (2016). Prohibitin overexpression improves myocardial function in diabetic cardiomyopathy. *Oncotarget* 7: 66–80.

- [6] Yu, Y. D., Xue, Y. T., & Li, Y. (2023). Identification and verification of feature biomarkers associated in heart failure by bioinformatics analysis. *Scientific Reports*, 13(1), 3488.
- [7] Brecker, M., Khakhina, S., Schubert, T. J., Thompson, Z., & Rubenstein, R. C. (2020). The probable, possible, and novel functions of ERp29. *Front Physiol* 11: 574339.
- [8] Lalonde, S., Codina-Fauteux, V. A., de Bellefon, S. M., Leblanc, F., Beaudoin, M., Simon, M. M., ... & Lettre, G. (2019). Integrative analysis of vascular endothelial cell genomic features identifies AIDA as a coronary artery disease candidate gene. *Genome Biology*, 20, 1-13.
- [9] Castillo-Avila, R. G., Gonzalez-Castro, T. B., Tovilla-Zarate, C. A., Martinez-Magana, J. J., Lopez-Narvaez, M. L., Juarez-Rojop, I. E., ... & Rodriguez-Perez, J. M. (2023). Association between genetic variants of CELSR2-PSRC1-SORT1 and cardiovascular diseases: a systematic review and meta-analysis. *Journal of Cardiovascular Development and Disease*, 10(3), 91.
- [10] Joshua, J., Caswell, J., O'Sullivan, M. L., Wood, G., & Fonfara, S. (2023). Feline myocardial transcriptome in health and in hypertrophic cardiomyopathy—A translational animal model for human disease. *PLoS One*, 18(3), e0283244.
- [11] Li, X., Huang, Y., Yin, D., Wang, D., Xu, C., Wang, F., ... & Wang, Q. K. (2013). Meta-analysis identifies robust association between SNP rs17465637 in MIA3 on chromosome 1q41 and coronary artery disease. *Atherosclerosis*, 231(1), 136-140.
- [12] Aggarwal, S., Narang, R., Saluja, D., & Srivastava, K. (2024). Diagnostic potential of SORT1 gene in coronary artery disease. *Gene*, 148308.
- [13] Nordestgaard, B. G., & Langsted, A. (2016). Lipoprotein (a) as a cause of cardiovascular disease: insights from epidemiology, genetics, and biology. *Journal of Lipid Research*, 57(11), 1953-1975.
- [14] Enas E A, Varkey B, Dharmarajan T S, et al. Lipoprotein (a): An independent, genetic, and causal factor for cardiovascular disease and acute myocardial infarction[J]. *Indian Heart Journal*, 2019, 71(2): 99-112.
- [15] Paquette, M., Dufour, R., & Baass, A. (2018). PHACTR1 genotype predicts coronary artery disease in patients with familial hypercholesterolemia. *Journal of Clinical Lipidology*, 12(4), 966-971.
- [16] Yuan, W., Zhang, W., Zhang, W., Ruan, Z. B., Zhu, L., Liu, Y., ... & Zhang, L. F. (2020). New findings in the roles of Cyclin-dependent Kinase inhibitors 2B Antisense RNA 1 (CDKN2B-AS1) rs1333049 G/C and rs4977574 A/G variants on the risk to coronary heart disease. *Bioengineered*, 11(1), 1084-1098.
- [17] Lek, M., Karczewski, K. J., Minikel, E. V., Samocha, K. E., Banks, E., Fennell, T., ... & Exome Aggregation Consortium. (2016). Analysis of protein-coding genetic variation in 60,706 humans. *Nature*, 536(7616), 285-291.
- [18] Ozaki, K., Sato, H., Inoue, K., Tsunoda, T., Sakata, Y., Mizuno, H., ... & Tanaka, T. (2009). SNPs in BRAP associated with risk of myocardial infarction in Asian populations. *Nature Genetics*, 41(3), 329-333.
- [19] Hinohara, K., Ohtani, H., Nakajima, T., Sasaoka, T., Sawabe, M., Lee, B. S., ... & Kimura, A. (2009). Validation of eight genetic risk factors in East Asian populations replicated the association of BRAP with coronary artery disease. *Journal of human genetics*, 54(11), 642-646.
- [20] Brecker, M., Khakhina, S., Schubert, T. J., Thompson, Z., & Rubenstein, R. C. (2020). The probable, possible, and novel functions of ERp29. *Frontiers in Physiology*, 11, 574339.

- [21] Karamanavi, E., McVey, D. G., van der Laan, S. W., Stanczyk, P. J., Morris, G. E., Wang, Y., ... & Ye, S. (2022). The FES gene at the 15q26 coronary-artery-disease locus inhibits atherosclerosis. *Circulation Research*, 131(12), 1004-1017.
- [22] Dong, W. Q., Chao, M., Lu, Q. H., Chai, W. L., Zhang, W., Chen, X. Y., ... & Zhang, M. X. (2016). Prohibitin overexpression improves myocardial function in diabetic cardiomyopathy. *Oncotarget*, 7(1), 66.
- [23] Ken-Dror, G., Talmud, P. J., Humphries, S. E., & Drenos, F. (2010). APOE/C1/C4/C2 gene cluster genotypes, haplotypes and lipid levels in prospective coronary heart disease risk among UK healthy men. *Molecular Medicine*, 16, 389-399.
- [24] Yu, Y. D., Xue, Y. T., & Li, Y. (2023). Identification and verification of feature biomarkers associated in heart failure by bioinformatics analysis. *Scientific Reports*, 13(1), 3488.

Responses to the comments of Reviewer #3

Comments to the Author

In this manuscript, Song and colleagues proposed a novel method to build prediction models of gene expressions in the context of multi-tissue datasets like GTEx. Via the analysis on 47 tissues in GTEx v8, the authors show that the proposed models outperform existing widely used methods: elastic nets (in PrediXcan) and UTMOST (a cross-tissue method).

As a method paper, it presented the method and evaluated the performance nicely but the problem that the method is trying to tackle is slightly outdated in the sense that the field has moved towards single-cell data so a bulk data-based approach is fundamentally less interesting to people. To make this work more impactful, it'd be great if the authors could run and share their models on other multi-tissue/multi-cell type datasets such as PsychENCODE or DICE. Besides, showcasing the ability to process single-cell data would be of great significance.

Thank you very much for your thoughtful and insightful comments. Based on your suggestions, we have applied our method on the DICE dataset^[1], which includes the transcriptomic information of 91 samples on 13 types of immune cells and 2 activation conditions. We cannot test our method on the PsychENCODE data, as the dataset is moved to the NIMH Data Archive, and cannot be accessed at this moment.

We first compared the performance of MTWAS, PrediXcan, TIGAR, and UTMOST with 13 types of immune cells and 2 activation conditions in the DICE dataset. We omitted the results of TIGAR because the inadequate sample size may lead to an over-fitted model for the method. MTWAS showed consistent improvement of prediction accuracy compared with PrediXcan and UTMOST, with an average improvement of 77.69% (SD=2.09%) and 5.87% (SD=1.17%) with regard to the prediction R^2 (Figure 1a below). MTWAS also identified about 5 times as many predictable genes as PrediXcan, and about 14 times as UTMOST (Figure 1b below).

We show that our method is also applicable to the single-cell RNA-seq (scRNA-seq) data. We generated pseudo-bulk data matrices comprising 14 immune cell types from the OneK1K dataset^[2], which consists of 1.27 million peripheral blood mononuclear cells (PMBCs) collected from 982 donors. Consistent with the results obtained from the bulk data, we found that MTWAS achieved the highest prediction R^2 , along with identifying the largest number of predictable genes across all cell types (Figure 2 below). Specifically, MTWAS identified from 84 (CD4 SOX4 cells) to 3,665 (NK cells) predictable genes under the stringent criterion, with an improvement of 104.8% (SD=61.6%) and 73.3% (SD=12.1%) compared with PrediXcan and UTMOST.

We have added the results of the DICE and OneK1K datasets in Lines 159-174 on Page 9.

Figure 1. The prediction accuracy on 13 types of immune cells and 2 activation conditions of the DICE dataset. a, The improvement of prediction R^2 over PrediXcan, of MTWAS and UTMOST. The prediction R^2 is based on 5-fold CV. **b,** The number of predictable genes by the three methods (FDR<0.05).

Figure 2. The prediction accuracy on 14 immune cell types of the OneK1K dataset. a, The improvement of prediction R^2 over PrediXcan, of MTWAS and UTMOST. The prediction R^2 is based on 5-fold CV. **b,** The number of predictable genes by the three methods (FDR<0.05). The cell types are arranged in ascending order based on their proportions of drop-out, from the lowest (CD4 NC) to the highest (CD4 SOX4).

Major comments:

1. For a tissue-specific expression, how well the imputation will perform? As it leverages info from other tissues to make predictions, would it only capture the tissue-shared component of the expression?

Thanks for your question. We agree with you that leveraging information from other tissues to make the imputation may dilute the signals of some tissue-specific eQTLs, which could be an issue for all multi-tissue methods. We used the imputed data to identify ct-eQTLs and ts-eQTLs. To find a balance between benefits gained from increasing sample sizes through imputation and possible drawbacks associated with the potential dilution of effects, we performed a weighted linear regression to estimate the effect sizes of eQTLs. Specifically, for a tissue with an observed sample size N_{obs} and an imputed sample size N_{imp} , we assign a weight of $\min(1, N_{obs}/N_{imp})$ to the imputed samples, and a weight of 1 to the observed samples. This weighting scheme prioritizes the impact of the observed data, especially when the imputed data volume surpasses the observed data, so as to retain tissue-specific signals in estimating eQTL effects.

We clarify that, throughout all of our experiments, the test data are the original observed data without being imputed (i.e., tissue-specific expression). We can see that MTWAS achieved the best performance among all the tested methods for GTEx, DICE, as well as the independent replication GEUVADIS studies. We also compared performance of PrediXcan, UTMOST, and TIGAR trained on the original datasets with those trained on the imputed datasets in GTEx. We observe that using imputed expression data for training consistently improves the prediction accuracy across all tissues for all methods. We have added a description of the weighting scheme in Lines 88-89 on Page 3, and Lines 354-359 on Pages 15 and 17.

2. In Figure 3 Uterus panel, maybe I'm not understanding it correctly but it seems strange to me that Uterus has more imputable genes than Muscle Skeletal even in PrediXcan. It seems that something is off. Similarly, it is unexpected to see small-sample-size tissues having a lot more imputable genes than tissue with a bigger sample size in Figure S2.

Thanks for your sharp observation and question. Figure 3 in the original manuscript provides two examples with the highest and lowest sample sizes in our GTEx study. The predictable genes were defined as genes with prediction R^2 larger than 0.01, which is a common cutoff widely used in TWAS studies such as PrediXcan^[3]. With this cutoff, we indeed can see more predictable genes in tissues with smaller sample sizes. This phenomenon has also been observed in Parrish et al. (2022)^[4].

A possible explanation is that, because of the small-sample-sizes, the sample variance of the null distribution of prediction R^2 in these tissues tend to be larger than that in tissues with larger sample sizes, resulting in more genes (in small-sample-size tissues) having their R^2 passing the 0.01 threshold, of which many are just false positives. We show a toy example by simulating two uncorrelated vectors under varying sample sizes, and summarize the proportion of cases where $R^2 > 0.01$ over 10,000 replicates (see Figure 3 below).

Because choosing 0.01 as a cutoff for the prediction R^2 for uterus and some other small-sample-size tissues may be too loose, we further introduce a more stringent criterion. Specifically, we utilize an F-statistic to test the significance of the predictability, and use a false discovery rate (FDR) cutoff at 0.05, which was also used in UTMOST paper^[5]. We have provided a figure showing the number of predictable genes with $FDR < 0.05$, and large-sample-size tissues have more predictable genes (Figure 4 below). We have also added a description of the stringent criterion of predictability in Lines 424-427 on Page 19.

Figure 3. The proportion of $R^2 \geq 0.01$ in null distribution with varying sample sizes.

Figure 4. The number of predictable genes of MWAS, UTMOST, PrediXcan, and TIGAR. The predictability with $FDR < 0.05$. The tissues are arranged in descending order based on their respective sample sizes, from the highest (muscle skeletal) to the lowest (uterus).

3. For the *RNF186* association in UC, the reported discovery is in the ovary which does not sound related to UC. Did you observe consistent results (sign or nominal significance) in more relevant tissue?

Thanks for point out this issue. Although we found that *RNF186* indeed plays an important role in ulcerative colitis^[6] ^[7] and colorectal cancer^[8], its expression in relevant tissues (e.g., colon transverse ($p = 0.13$) and colon sigmoid ($p = 0.56$)) in our analysis did not pass the significance level. It may be because of the limited sample sizes. We decided to remove the elaboration on this association from our manuscript to avoid misleading the reader.

4. Considering EBIC as a variable selection approach, how does it perform compared to a simpler approach like forward selection or a fine-mapping approach like SuSiE?

Thanks for the question. As discussed in Li and Liu (2018), under high-dimensional settings, the forward step ensures the screening consistency, and the backward step ensures the individual term

selection consistency^[9]. The combination of both establishes the model selection consistency. It has also been revealed that the backward selection procedure can reduce the overfitting of the data^[10], such as the multicollinearity across variables. Considering the implementation, the backward step eliminates variables from the subsets selected by the forward step, so it does not introduce any serious additional computational burden. Therefore, we leverage the forward-backward method to optimize EBIC.

As for the fine-mapping approach, we first clarify that our method aims at improving the performance of transcriptome-wide association studies (TWAS), which differs significantly from the aim of fine-mapping, such as SuSiE^[11]. More specifically, TWAS methods aim at identifying the genes associated with a phenotype of interest, whereas those fine-mapping methods aim at identifying causal variants, which makes inferences on whether the effect size for a specific SNP is non-zero, or the probability of a specific SNP set covers a causal SNP^{[12][13]} to the phenotype.

It is true that if we regard the expression levels as a certain phenotype, the fine-mapping methods are based on the same linear model as that used in the prediction stage of TWAS. The difference lies in that the goal of TWAS is to find the best predictor and maximize the prediction accuracy of gene expression levels. In contrast, the goal of fine-mapping methods is to narrow down the list of potential causal variants within a genomic region^{[12][13]}.

Taking SuSiE as an example, in order to construct the credible set, it needs to impose some more assumptions on the distribution of the effect sizes, such as the effect sizes $\beta = \sum_{l=1}^L \beta_l$, $\beta_l = \beta_l \gamma_l$, $\gamma_l \sim Mult(1, \pi)$, $\beta_l \sim N(0, \sigma_{0l}^2)$. The number of causal variants (L) also needs to be specified. In contrast, our framework does not require these assumptions. Therefore, when the assumptions match the data structure, SuSiE will have a good performance. Otherwise, it may perform badly. We illustrate this point using the whole blood tissue in GTEx dataset. From Table 1 below we can see that, although the mean prediction R^2 of SuSiE is higher than UTMOST, the number of predictable genes of SuSiE is smaller than that of UTMOST. This is likely because SuSiE performed outstandingly well on some specific gene sets, but performed substantially worse on many others.

Nevertheless, we believe that it is a promising direction to integrate fine-mapping methods with TWAS frameworks to prioritize causal genes for targeted diseases, and we wish to pursue this direction in our future research. We have added some discussions in Lines 304-309 on Page 14.

Table 1. The prediction accuracy of the TWAS methods MTWAS, UTMOST, PrediXcan, TIGAR, and the fine-mapping method SuSiE on GTEx whole blood tissue. The results are based on 5-fold CV. The highest prediction R^2 and the largest number of predictable genes are highlighted in boldface.

Method	MTWAS	UTMOST	PrediXcan	TIGAR	SuSiE
Mean prediction R^2	0.041	0.036	0.034	0.038	0.037
Mean signed Prediction R^2	0.037	0.031	0.031	0.033	0.034
# Predictable genes ($R^2 > 0.01$)	10,444	10,002	6,991	9,601	7,326
# Predictable genes (FDR < 0.05)	6,391	5,161	4,534	5,409	5,204

Minor comments:

1. At line 70 “Then, ...”, the first part of this sentence (“we impute the missing entries ..”) is confusing to me since the previous sentence has mentioned the imputation step so it sounds duplicated.

Thanks for pointing it out and sorry for the confusion. We have removed this sentence to make it clearer.

2. At line 77 “We use ...”, this “2” sounds like a GTEx-specific parameter? As in other data sets, we may not have ~50 tissues to begin with so that the number should be changed? It might be worth to make this clearer and more generic rather than presented as a magic number.

Thanks for your suggestion. The number of PCs used to identify cross-tissue eQTLs is decided by the eigenvalues of the tissue correlation matrix. Intuitively, an eigenvalue of 2 means that the principal component would explain about two variables’ worth of the variability. The rationale for using 2 as the default cutoff for the eigenvalues is that each component would explain at least two tissues’ worth of the variability, which leads to about 5 PCs on average. Therefore, “2” is not a GTEx-specific parameter, while “5” could be. When we consider fewer tissues, using cutoff 2 to the eigenvalues may lead to a smaller number of PCs. We have added some clarifications in Lines 337-339 on Page 15, and implemented an automatic selection of the PC number in our software.

3. At line 81, it is unclear what “tissue-wise” means here. From my understanding, this step is done one tissue at a time (Step 3 in Figure 1) so this is not tissue-wise.

Thanks for pointing it out. Yes, we mean doing one tissue at a time. We have corrected this expression in the revised manuscript. We copy it here for easy review:

“Specifically, with the tissue-specific gene expression as the response and genotypes of the cis-SNP as the predictors in a linear model, SODA selects ts-eQTLs by minimizing EBIC (after accounting for ct-eQTLs).” (Lines 86-88 on Page 3)

“For each tissue, we regress its each gene’s expression levels against genotypes of the gene’s cis-SNPs, and select variables based on the EBIC criterion.” (Lines 345-346 on Page 15)

4. When using R^2 for performance evaluation, It is better to consider the sign, i.e. signed R^2 .

Thanks for your suggestion. We agree that the signed prediction R^2 (the sign of Pearson’s correlation coefficient times the R^2) is important for performance evaluation. We have updated our results with signed prediction R^2 provided (Tables 2 and 3 below, and Tables 1 and 2 in the revised manuscript). As the signed prediction R^2 can be negative, it tends to be smaller and more conservative compared with the prediction R^2 (Tables 2 and 3 below). Figure 5(b) below shows the improvement of average

signed prediction R^2 over PrediXcan. We observed that the relative performances of TIGAR over PrediXcan could be opposite with regard to prediction R^2 or signed prediction R^2 (e.g., in liver and brain nucleus). Nevertheless, the superiority of MTWAS compared with the other methods remains unchanged evaluated by signed prediction R^2 .

Table 2. The prediction accuracy of MTWAS, PrediXcan, UTMOST, and TIGAR on 47 GTEx tissues. The highest Prediction R^2 and signed prediction R^2 are highlighted in boldface.

	MTWAS	PrediXcan	UTMOST	TIGAR
Average prediction R^2	0.061	0.041	0.056	0.044
Average signed prediction R^2	0.047	0.031	0.041	0.030

Table 3. Replication study on the GEUVADIS cohort for lymphoblastoid cell lines. The training weights are based on the Epstein-Barr virus transformed lymphocytes in the GTEx datasets. The highest prediction R^2 and signed prediction R^2 are highlighted in boldface.

	MTWAS	PrediXcan	UTMOST	TIGAR
Average prediction R^2	0.034	0.023	0.032	0.015
Average signed prediction R^2	0.031	0.022	0.030	0.013

Figure 5. Improvements of average (signed) prediction R^2 over PrediXcan, of MTWAS, MTWAS-tissue, UTMOST, and TIGAR, evaluated on 47 tissues of the GTEx datasets. The tissues are arranged in descending order based on their respective sample sizes, from the highest (muscle skeletal) to the lowest (uterus). **a**, The performance with regard to prediction R^2 . **b**, The performance with regard to signed prediction R^2 . The results are based on 5-fold CV.

5. In identifying ts-eQTL, what's the motivation for fixing the eQTLs identified in previous PCs for the analysis of the next PCs? It's not clear to me why the next PC needs to have the eQTLs

from all previous PCs as eQTLs.

Thanks for the question. In the PCA analysis, the former PC accounts for a larger proportion of the variance. Thus, including the previous eQTLs into the next PC model can be regarded as a form of penalization, to mitigate the risk of excessive variable selection as ct-eQTLs. We clarify that the mere presence of an eQTL in the model does not imply an unchanged effect on subsequent PCs. We allow for the possibility that effects of certain pre-fixed eQTLs are zero. We have added some explanation in Lines 335-336 on Page 15.

References:

- [1] Schmiedel, B. J., Singh, D., Madrigal, A., Valdovino-Gonzalez, A. G., White, B. M., Zapardiel-Gonzalo, J., ... & Vijayanand, P. (2018). Impact of genetic polymorphisms on human immune cell gene expression. *Cell*, 175(6), 1701-1715.
- [2] Yazar, S., Alquicira-Hernandez, J., Wing, K., Senabouth, A., Gordon, M. G., Andersen, S., ... & Powell, J. E. (2022). Single-cell eQTL mapping identifies cell type-specific genetic control of autoimmune disease. *Science*, 376(6589), eabf3041.
- [3] Gamazon, E. R., Wheeler, H. E., Shah, K. P., Mozaffari, S. V., Aquino-Michaels, K., Carroll, R. J., ... & Im, H. K. (2015). A gene-based association method for mapping traits using reference transcriptome data. *Nature Genetics*, 47(9), 1091-1098.
- [4] Parrish, R. L., Gibson, G. C., Epstein, M. P., & Yang, J. (2022). TIGAR-V2: Efficient TWAS tool with nonparametric Bayesian eQTL weights of 49 tissue types from GTEx V8. *Human Genetics and Genomics Advances*, 3(1).
- [5] Hu, Y., Li, M., Lu, Q., Weng, H., Wang, J., Zekavat, S. M., ... & Zhao, H. (2019). A statistical framework for cross-tissue transcriptome-wide association analysis. *Nature Genetics*, 51(3), 568-576.
- [6] Fujimoto, K., Kinoshita, M., Tanaka, H., Okuzaki, D., Shimada, Y., Kayama, H., ... & Takeda, K. (2017). Regulation of intestinal homeostasis by the ulcerative colitis-associated gene RNF186. *Mucosal Immunology*, 10(2), 446-459.
- [7] Rivas, M. A., Graham, D., Sulem, P., Stevens, C., Desch, A. N., Goyette, P., ... & Daly, M. J. (2016). A protein-truncating R179X variant in RNF186 confers protection against ulcerative colitis. *Nature Communications*, 7(1), 12342.
- [8] Ji, Y., Tu, X., Hu, X., Wang, Z., Gao, S., Zhang, Q., ... & Chen, W. (2020). The role and mechanism of action of RNF186 in colorectal cancer through negative regulation of NF- κ B. *Cellular Signalling*, 75, 109764.
- [9] Li, Y., & Liu, J. S. (2019). Robust variable and interaction selection for logistic regression and general index models. *Journal of the American Statistical Association*, 114(525), 271-286.
- [10] Hastie, T., Tibshirani, R., Friedman, J. H., & Friedman, J. H. (2009). *The elements of statistical learning: data mining, inference, and prediction* (Vol. 2, pp. 1-758). New York: springer.
- [11] Wang, G., Sarkar, A., Carbonetto, P., & Stephens, M. (2020). A simple new approach to variable selection in regression, with application to genetic fine mapping. *Journal of the Royal Statistical Society Series B: Statistical Methodology*, 82(5), 1273-1300.
- [12] Liang, Y., Aguet, F., Barbeira, A. N., Ardlie, K., & Im, H. K. (2021). A scalable unified

framework of total and allele-specific counts for cis-QTL, fine-mapping, and prediction. *Nature Communications*, 12(1), 1424.

- [13] Li, X., Sham, P. C., & Zhang, Y. D. (2024). A Bayesian fine-mapping model using a continuous global-local shrinkage prior with applications in prostate cancer analysis. *The American Journal of Human Genetics*, 111(2), 213-226.

REVIEWERS' COMMENTS

Reviewer #1 (Remarks to the Author):

The concerns have been answered, I am satisfied with the response.

Reviewer #2 (Remarks to the Author):

My comments are well addressed.

Reviewer #3 (Remarks to the Author):

The authors addressed all my comments thoroughly with clear explanation and ridge justifications and appropriate modifications have been reflected in the updated manuscript. With all these being said, I have only some minor comments and follow-up questions as below.

1. Regarding the OneK1K results. The performance improvement in OneK1K data is quite variable across cell types. Could you comment on the potential reasons for this? This might provide insights when users apply this approach to their datasets.
2. Regarding the weighting scheme for getting effect sizes for ct-QTLs and ts-QTLs. Out of curiosity, does this weighing scheme help the performance at all? Besides, if imputation is mostly capturing the tissue-shared part, do the imputed samples actually add noise rather than signal to ts-eQTL part? If so, it might be helpful to exclude them from ts-eQTL identification?
3. Just to follow up on integrating fine-mapping approach for prediction model training. I was thinking about approaches like <https://onlinelibrary.wiley.com/doi/full/10.1002/gepi.22346> where fine-mapping is served as a way to get “causal QTLs” for a gene. But anyway, I appreciate your thoughts and experiments along this line.

Reviewer #3 (Remarks on code availability):The described methods (both training new models and applying pre-trained models) are implemented in the provided GitHub repository as an R package. They provided in-depth tutorials on applying their package,

We thank the editor and associate editor for the decision to publish our paper in principle. We are very grateful to the referees for their dedications, thoughtful comments, and constructive suggestions.

Responses to the comments of Reviewer #3

The authors addressed all my comments thoroughly with clear explanation and ridge justifications and appropriate modifications have been reflected in the updated manuscript. With all these being said, I have only some minor comments and follow-up questions as below.

Thank you very much for the positive comment. We really appreciate your constructive comments and suggestions, which helped us improve the paper a lot.

1. Regarding the OneK1K results. The performance improvement in OneK1K data is quite variable across cell types. Could you comment on the potential reasons for this? This might provide insights when users apply this approach to their datasets.

Thanks for your question. Supplementary Figure 8a (Figure 1a below) shows the improvement of our method over PrediXcan and UTMOST on the OneK1K dataset. The cell types are arranged in descending order based on cell counts within each cell type, from the largest (CD4 NC, $N_{\text{cell}}=463,528$) to the smallest (Plasma, $N_{\text{cell}}=3,625$). The variation in cell counts mainly comes from both cell proportions and the number of individuals with identifiable cells^[1]. In general, all methods identified more predictable genes in cell types with more cell counts. Notably, the improvements of multi-cell-type TWAS methods (both MTWAS and UTMOST) over the single-cell-type method (PrediXcan) are more significant in cell types with fewer cell counts and lower statistical power in identifying predictable genes (Figure 1b below). This pattern aligns with observations from bulk studies where multi-tissue methods show more benefits in tissues with smaller sample sizes (see Figure 2a in the manuscript). This finding underscores the value of leveraging cross-tissue/cell-type information to improve prediction in cell types with limited data. We have added some comments in Lines 172-176 on Page 6.

Figure 1. The prediction accuracy on 14 immune cell types of the OneK1Kdataset. a The improvement of prediction R^2 over PrediXcan, of MTWAS and UTMOST. The prediction R^2 is based on 5-fold CV. **b** The number of predictable genes by the three methods (FDR<0.05). The cell types are arranged in descending order based on the total cell counts, from the largest (CD4 NC, $N_{\text{cell}}=463,528$) to the smallest (Plasma, $N_{\text{cell}}=3,625$).

2. Regarding the weighting scheme for getting effect sizes for *ct-QTLs* and *ts-QTLs*. Out of curiosity, does this weighing scheme help the performance at all? Besides, if imputation is mostly capturing the tissue-shared part, do the imputed samples actually add noise rather than signal to *ts-eQTL* part? If so, it might be helpful to exclude them from *ts-eQTL* identification?

Thank you for your questions and suggestions. The weighting scheme incorporates cross-tissue information and retains tissue-specific signals. Thus, it improves the prediction accuracy compared with estimating effect sizes using only observed samples (see Setting 1 below). As for the selection of *ts-eQTL*, we agree with you that the imputation is mostly capturing the tissue-shared part. However, the imputation also involves interaction and nonlinear information that may carry tissue-specific component, and our experiments (see Setting 2 below) show moderate benefits of imputed samples on tissue-specific eQTL selection. We have added some discussion in Lines 351-355, and Lines 365-368 on Pages 12-14.

For better illustration, we consider two additional settings for selecting eQTLs and estimating effect sizes:

*Setting 1: Using imputed data for identifying both *ct-eQTLs* and *ts-eQTLs*, while using only observed samples to estimate effect sizes;*

*Setting 2: Using imputed data for identifying *ct-eQTLs*, while using only observed samples to identify *ts-eQTL*, combined with a weighting scheme for estimating effect sizes.*

The comparison between our current model and Setting 1 demonstrates the benefits of the weighting scheme. We first applied Setting 1 to 47 tissues of the GTEx dataset, and found that our current model achieved significantly higher prediction R^2 than Setting 1 on 46 of the 47 tissues (one-sided

paired Wilcoxon test for R^2 , $p < 0.05$). We further performed a replication study by training weights with the GTEx samples for Epstein-Barr virus transformed lymphocytes, and applying them to the GEUVADIS lymphoblastoid cell lines for evaluation. As shown in Table 1 below (Supplementary Table 3 in the revised manuscript), our current weighting scheme outperforms Setting 1 in terms of both prediction R^2 and the number of predictable genes.

As for the ts-eQTL identification, we consider Setting 2 which selects ts-eQTLs with only observed samples. We found that our current model significantly outperforms Setting 2 in 20 out of 47 tissues, while Setting 2 has significantly higher prediction R^2 in only 5 out of 47 tissues (one-sided paired Wilcoxon test for R^2 , $p < 0.05$). Additionally, our model showed better performance in the replication study regarding both prediction R^2 and the number of predictable genes (Table 1 below).

Table 1. The performance of MTWAS and other two settings in a replication study on the GEUVADIS cohort for lymphoblastoid cell lines. The training weights are based on the EBV transformed lymphocytes in the GTEx datasets. The highest prediction R^2 and signed prediction R^2 , and the largest number of predictable genes under the common and stringent criteria are highlighted in boldface.

	MTWAS (current weighting scheme)	Setting 1	Setting 2
Average prediction R^2	0.034	0.033	0.032
Average signed prediction R^2	0.031	0.031	0.028
# predictable genes ($R^2 > 0.01$)	5,176	5,157	5,041
# predictable genes ($FDR < 0.05$)	4,339	4,322	4,171

3. Just to follow up on integrating fine-mapping approach for prediction model training. I was thinking about approaches like <https://onlinelibrary.wiley.com/doi/full/10.1002/gepi.22346> where fine-mapping is served as a way to get “causal QTLs” for a gene. But anyway, I appreciate your thoughts and experiments along this line.

Thank you very much for your insightful comments. In our previous response, we included SuSiE as a method to identify “causal QTLs” for a gene and compared it with our method in the prediction of expression levels (Table 2 below, and Supplementary Table 2 in the manuscript). Additionally, we have also added some discussions on integrating causal QTLs and cross-tissue information in TWAS studies (Lines 307-311 on Page 11). We appreciate your valuable suggestions and agree that it is a promising direction. We intend to pursue this direction in our future research.

We copy our discussion here for easy review:

“We note that leveraging some fine-mapping methods, such as SuSiE^[2] and DAP-K^[3] ^[4] for identifying causal QTLs, can improve the prediction accuracy for certain genes (Supplementary Table 2). Therefore, it is conceptually advantageous to integrate fine-mapping methods into our cross-tissue TWAS framework to prioritize causal genes, which can be a promising future direction of our research.”

Table 2. The prediction accuracy of the TWAS methods MTWAS, UTMOST, PrediXcan, TIGAR, and the fine-mapping method SuSiE on GTEx whole blood tissue. The results are based on 5-fold CV. The highest prediction R^2 and the largest number of predictable genes are highlighted in boldface.

Method	MTWAS	UTMOST	PrediXcan	TIGAR	SuSiE
Mean prediction R^2	0.041	0.036	0.034	0.038	0.037
Mean signed Prediction R^2	0.037	0.031	0.031	0.033	0.034
# Predictable genes ($R^2 > 0.01$)	10,444	10,002	6,991	9,601	7,326
# Predictable genes ($FDR < 0.05$)	6,391	5,161	4,534	5,409	5,204

Reviewer #3 (Remarks on code availability):

The described methods (both training new models and applying pre-trained models) are implemented in the provided GitHub repository as an R package. They provided in-depth tutorials on applying their package.

Thank you very much for your positive comments. We greatly appreciate your inspiring and constructive comments and suggestions.

References:

- [1] Yazar, S., Alquicira-Hernandez, J., Wing, K., Senabouth, A., Gordon, M. G., Andersen, S., ... & Powell, J. E. (2022). Single-cell eQTL mapping identifies cell type-specific genetic control of autoimmune disease. *Science*, 376(6589), eabf3041.
- [2] Wang, G., Sarkar, A., Carbonetto, P., & Stephens, M. (2020). A simple new approach to variable selection in regression, with application to genetic fine mapping. *Journal of the Royal Statistical Society Series B: Statistical Methodology*, 82(5), 1273-1300.
- [3] Wen, X., Lee, Y., Luca, F., & Pique-Regi, R. (2016). Efficient integrative multi-SNP association analysis via deterministic approximation of posteriors. *The American Journal of Human Genetics*, 98(6), 1114-1129.
- [4] Barbeira, A. N., Melia, O. J., Liang, Y., Bonazzola, R., Wang, G., Wheeler, H. E., ... & Im, H. K. (2020). Fine-mapping and QTL tissue-sharing information improves the reliability of causal gene identification. *Genetic Epidemiology*, 44(8), 854-867.